# GeoLoom: High-quality Geometric Diagram Generation from Textual Input

## Abstract

High-quality geometric diagram generation presents both a challenge and an opportunity: it demands strict spatial accuracy while offering well-defined constraints to guide generation. Inspired by recent advances in geometry problem solving that employ formal languages and symbolic solvers for enhanced correctness and interpretability, we propose GeoLoom, a novel framework for text-to-diagram generation in geometric domains. GeoLoom comprises two core components: an autoformalization module that translates natural language into a specifically designed generation-oriented formal language GeoLingua, and a coordinate solver that maps formal constraints to precise coordinates using the efficient Monte Carlo optimization. To support this framework, we introduce GeoNF, a dataset aligning natural language geometric descriptions with formal GeoLingua descriptions. We further propose a constraint-based evaluation metric that quantifies structural deviation, offering mathematically grounded supervision for iterative refinement. Empirical results demonstrate that GeoLoom significantly outperforms state-of-the-art baselines in structural fidelity, providing a principled foundation for interpretable and scalable diagram generation.

## 1 Introduction

Geometric diagrams play a crucial role in mathematical education, scientific research, and engineering design, serving as the core medium for concept visualization and knowledge transfer. Despite their significance, diagram generation remains predominantly manual and labor-intensive (Zhang et al., 2007; Yu et al., 2015). Conventional methods, primarily rule-based or interactive tools (Yu et al., 2015), depend heavily on user input and domain expertise, resulting in limited efficiency and scalability. These approaches typically require explicit specification of geometric primitives and constraints, rendering them inadequate for large-scale content creation. This highlights that automated models are urgently needed to convert natural language into accurate geometric diagrams.

In recent years, text-to-image generation models (Zhang et al., 2023a; Jia et al., 2024), such as Stable Diffusion (Rombach et al., 2022), DALL·E (Ramesh et al., 2022), and their successors (Jiahui et al., 2022; Chang et al., 2023; Saharia et al., 2022), have achieved significant progress in synthesizing high-fidelity natural images from textual prompts. However, these models remain inadequate for generating geometric diagrams, primarily due to the sparse and highly structured nature of such visuals. Even with domain-specific fine-tuning, current models exhibit poor spatial accuracy, rendering the outputs unsuitable for educational contexts. A primary limitation is the absence of high-quality paired datasets; although resources like Geometry3K (Lu et al., 2021), GeoQA (Chen et al., 2021) and PGPS9K (Zhang et al., 2023b) facilitate geometric understanding, they offer limited supervision for generative tasks as they lack well-aligned textual and diagrammatic data.

Unlike natural images, which are often evaluated based on subjective perceptual quality (Radford et al., 2021), geometric diagrams are inherently sparse, demanding exact placement of points, lines, and angles, and any distortion undermines their mathematical validity (Chen et al., 2022). This divergence presents both a challenge and an opportunity. The challenge lies in capturing precise spatial relationships, logical rules, and symbolic fidelity, capabilities that current models lack. Conversely, the well-defined nature of geometric diagrams provides explicit evaluation metrics, offering a principled foundation for supervising the generation process. Thus, constraints of geometric dia-

Figure 1: An overview of the analogy between geometry problem solving and geometric diagram generation, highlighting the central role of formal language in ensuring geometric correctness.

gram generation provides objective, domain-specific criteria for evaluation, that can supervise and calibrate generation quality in a principled manner.

In this paper, we introduce GeoLingua, a novel formal language for text-to-diagram generation in geometric domains. Central to GeoLingua is the use of a formal language as an intermediate representation to encode geometric entities and relations, enabling precise computation of point coordinates necessary for accurate diagram rendering. This design draws on established practices in geometry problem solving, where formal languages coupled with symbolic solvers ensure logical consistency and computational rigor through structured, unambiguous semantics (Chervonyi et al., 2025). Figure 1 shows this analogy. However, existing formal languages are ill-suited for diagram synthesis due to the absence of explicit constructive dependencies. To overcome this limitation, we propose GeoLingua, a generative-oriented formal language that encodes both geometric constraints and point dependencies, explicitly specifying construction order and topological structure. Building on GeoLingua, we construct GeoNF, a paired dataset comprising aligned natural language descriptions and their formal representations, serving as a benchmark for training and evaluating text-to-diagram generation models in geometric domains.

To this end, GeoLoom is designed as a two-stage framework comprising an **autoformalization module** and a **coordinate solver**, which work in tandem to enable precise translation from linguistic descriptions to structured diagrams. 1) The autoformalization module translates natural language into a structured formal representation using GeoLingua, a syntactically and semantically tailored formal language designed to express geometric constraints and constructive dependencies. GeoLingua organizes geometric content into two fundamental categories: geometric primitives and constraints. Primitives include free and dependent points, reflecting their constructional hierarchy, while constraints encode spatial relations, such as lengths and angles, that define the topological relations among primitives. 2) The second module, the coordinate solver, optimizes spatial coordinates by satisfying geometric constraints for diagram rendering. To solve it efficiently, we adopt a Monte Carlo-based optimization algorithm (Metropolis & Ulam, 1949), which iteratively refines the positions of free points through stochastic perturbations. During optimization, we introduce a quantitative objective that evaluates the extent to which geometric constraints are violated, and then seek to minimize these violations by probabilistically sampling from the solution space.

Experimental results show that our proposed model, GeoLoom, achieves significant improvements over state-of-the-art baselines in both qualitative and quantitative evaluations. Furthermore, analysis of generation time demonstrates superior computational efficiency, supporting practical deployment in educational settings. In summary our contributions are,

- We design GeoLingua, a generation-oriented formal language that encodes geometric primitives, constraints, and constructive dependencies. Based on this, we construct GeoNF, a high-quality paired dataset aligning natural language descriptions with formal representations, providing a robust benchmark for GeoLingua autoformalization evaluation.

- We propose GeoLoom, a two-stage framework that integrates structured autoformalization with constraint-guided coordinate solving. The solver employs Monte Carlo-based optimization to enforce spatial accuracy under geometric constraints.

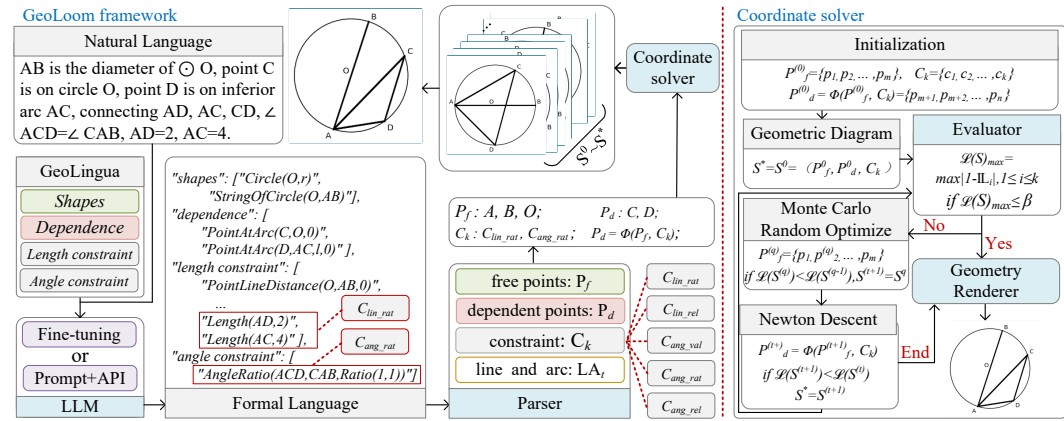

Figure 2: Illustrating GeoLoom framework: an autoformalization module and a coordinate solver.

- Empirically we demonstrate that GeoLoom achieves state-of-the-art performance with high efficiency, supporting practical deployment in educational scenarios.

## 2 GEOLINGUA

While formal languages have long supported geometry solving, existing representations often lack the capacity to capture topological dependencies essential for accurate diagram synthesis (Polu & Sutskever, 2020; Lu et al., 2021). This section introduces GeoLingua, a generation-oriented formal language designed for geometric constructions. To ensure its rigor, we conducted systematic consultations with mathematics experts. We first formally define the syntax and semantics of GeoLingua and then present GeoNF, a benchmark dataset comprising aligned natural language descriptions, formal representations, and diagrams to facilitate supervised training and evaluation.

**Syntax and Semantics.** We conduct a systematic analysis of the underlying compositional structure of geometry problems. This analysis reveals four core components that underpin the generation of geometric diagrams: (1) Shapes, (2) Dependence, (3) length constraints, and (4) Angle constraints. Building upon this abstraction, we introduce GeoLingua, a formal language framework designed to translate natural language (NL) problem descriptions into structured formal language (FL) representations. These representations are explicitly organized along the four axes mentioned above, enabling downstream coordinate optimization. We present an overview of this representational schema and the detailed definition can be found in Appendix A.

Shapes: The foundational elements of a diagram are basic geometric primitives such as line segments, triangles, and circles. The defining vertices of these primitives are categorized as free points $P_f$, whose coordinates can be independently determined (e.g., the two endpoints $A$ and $B$ of the $AB$ line segment);

Dependence: Dependent points $P_d$ are determined based on free points and geometric constraints (e.g., midpoints, perpendicular, intersections);

Length constraint: These include length ratios $C_{lin\_rat}$ and length relations $C_{lin\_rel}$, encoding proportional and relational constraints between line segments;

Angle constraint: These encompass fixed angle value $C_{ang\_val}$, angular ratios $C_{ang\_rat}$, and angular relations $C_{ang\_rel}$, specifying both absolute and relative angular properties.

**GeoNF Dataset.** Building upon the developed formal language framework, we present GeoNF, a curated dataset of geometry problems sourced from national educational examinations, specifically selected for their amenability to geometric diagram construction. To annotate the dataset, we recruited mathematics majors who were trained to translate natural language problem statements into structured formal representations. Each annotation underwent manual verification to ensure semantic fidelity and syntactic correctness.

GeoNF involves various types of geometric relations and geometric shapes, and we classify and statistically analyze them according to these relations and geometric shapes. The specific distribution is shown in the Figure 3. The resulting GeoNF comprises 4,730 high-quality aligned pairs of natural and formal language representations, denoted as $\{N_n, F_n\}$, where $N_n$ represents the natural language description of a geometric problem, and $F_n$ denotes its corresponding formal expression. These formal expressions encode geometric entities, spatial relations, and constraint conditions necessary for downstream geometric computation and diagram rendering. The dataset is partitioned into 4,300 training pairs and 430 test pairs, enabling models to learn mappings from informal language to formal specifications, thereby supporting constraint understanding.

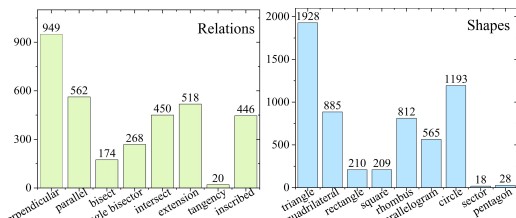

Figure 3: Geometric relations and geometric shape distributions in the GeoNF dataset.

## 3 GEOLOOM

Based on GeoLingua, we propose GeoLoom, a two-stage framework for high-precision text-to-diagram generation in geometry domains. GeoLoom bridges natural language descriptions and structured geometric diagrams by integrating an autoformalization module with a coordinate solver. This section provides a detailed overview of GeoLoom, outlining its design rationale, modular architecture, and the collaborative operation of its components. An overview is illustrated in Figure 2.

### 3.1 AUTOFORMALIZATION

We investigate two complementary approaches: (1) a training-free, prompt-based method, and (2) a fine-tuning-based adaptation of LLMs. We have set up validation-based filter to address the issue of illusion in large models, ensuring that the output formal language only contains the formal statements we define. In both settings, the generated formal expressions are subsequently processed via Monte Carlo optimization to compute precise coordinates, further enabling high-quality generation.

**Training-free.** We develop a prompt-engineering pipeline that interfaces directly with LLMs (DeepSeeek-V3 (Liu et al., 2024)), enabling translation from natural language to structured formal representations without additional training. This pipeline integrates a validation mechanism to ensure syntactic and semantic compliance with the target formal language. Using the GeoNF test set, the process unfolds in three stages: (i) prompt construction guided by geometric semantics, (ii) structured generation of candidate formal expressions, and (iii) validation-based filtering (prompt and detailed validation methods can be found in Appendix B).

**Fine-tuning.** To enhance model adaptability for formal language generation in geometric contexts, we conduct fine-tuning experiments across different model families (LLaMA (Grattafiori et al., 2024) and Qwen (Bai et al., 2023)) and different parameter scales. Each model is fine-tuned on the training split of the GeoNF dataset, which contains aligned pairs of natural language inputs and corresponding formal representations.

This dual-path investigation offers a comprehensive comparison of zero-shot and adaptation-based strategies for autoformalization, establishing a foundation for robust formal language generation in geometry-centric tasks.

### 3.2 COORDINATE SOLVER

**Geometric Constraint Deviation.** The well-structured nature of geometric diagrams enables the use of explicit, quantifiable evaluation metrics, thereby providing a principled basis for supervising the diagram generation process. While general image evaluation (Lin et al., 2024) typically focuses on the presence of basic visual features and semantic alignment with textual input (Radford et al., 2021), the evaluation of geometric diagrams necessitates a stricter adherence to mathematical precision. Specifically, this includes the accurate placement of point coordinates and compliance with geometric constraints related to line segments and angles. We hence identify five fundamental types of geometric constraints that govern the structural fidelity of diagrams and formalize each as follows:

| Natural Language | Origin-img | Ours (training-free) | Ours (fine-tuned) | AutomaTikZ | Seedream |
|---|---|---|---|---|---|
| The triangle ABC is an inscribed equilateral triangle of circle O, point O is the center, po-ints D and E are on sides AC and AB respec-tively, connecting OD and OE, DA=BE. | | | | | |
| In triangle ABC, points D, E, and F are res-pectively the midpoints of AB, AC, and BC, connecting DE, EF, and DF. Points P, M, and N are respectively the midpoints of DE, DF, and EF, con-necting PM, PN, and MN. | | | | | |
| I n the four sides ABCD, ∠BAD=120◦, ∠ABC=∠ADC=90◦, find points M and N on BC and CD respectively, and connect AM and AN. | | | | | |
| In rectangle ABCD, point E is on edge AB, connected to DE, which is the bisector of ∠ADC. Point F is on the extension of DE, connected to BF, ∠BFE = 90◦, connected to AF, CF, CF, and AB intersecting with G. | | | | | |

Figure 4: Qualitative comparison with AutomaTikz (based on LLaMa7b) and SeeDream model. (The corresponding formal language and the more results can be found in Appendix D.)

- **Length ratio.** Given a target length ratio $L_i^t/L_j^t = R_{tar\_line}$, the length ratio for generated diagram is $L_i^g/L_j^g = R_{gen\_line}$. The evaluation metric for this constraint is defined as:

$$\mathcal{C}_{lin\_rat} = R_{gen\_line}/R_{tar\_line} \tag{1}$$

- **Length relation.** For a relational expression between two segments $L_{left} \odot L_{right}$, where $\odot \in \{=, >, \geq, <, \leq\}$, the evaluation is given by:

$$\mathcal{C}_{lin\_rel} = [\odot = \text{" = "}] \cdot L_{left}/L_{right} + [\odot = \text{" } \star \text{ "} \wedge L_{left} \star L_{right}] \tag{2}$$

where $\star$ can be $> (\geq)$ or $< (\leq)$, $L_{left}$ represents the left-hand side of the length relational expression, and $L_{right}$ represents the right-hand side of the length relational expression.

- **Angle value.** Given a target angle $\theta_t$ and a generated angle $\theta$, the constraint evaluation is:

$$\mathcal{C}_{ang\_val} = [\theta_t \neq 0 \wedge \theta \neq 0] \cdot \theta/\theta_t + [\theta_t = 0 \wedge \theta = 0] \tag{3}$$

- **Angle ratio.** Given a target angle ratio $\theta_i^t/\theta_j^t = R_{tar\_angle}$, and a generated ratio $\theta_i^g/\theta_j^g = R_{gen\_angle}$, the constraint metric is:

$$\mathcal{C}_{ang\_rat} = R_{gen\_angle}/R_{tar\_angle} \tag{4}$$

- **Angle relation.** For an angle relation $A_{left} \odot A_{right}$, with $\odot \in \{=, >, \geq, <, \leq\}$, the corresponding evaluation is:

$$\mathcal{C}_{ang\_rel} = [\odot = \text{" = "}] \cdot A_{left}/A_{right} + [\odot = \text{" } \star \text{ "} \wedge A_{left} \star A_{right}] \tag{5}$$

where $\star$ can be $> (\geq)$ or $< (\leq)$, $A_{left}$ ($A_{right}$) represents the left-hand (right-hand) side of the angle relational expression. (In the above equation, [] is an Iverson bracket.)

The above five constraint values represent the set of constraint satisfaction scores, normalized to the interval $[0, 1]$, where a value of 1 means that the corresponding geometric constraint is perfectly satisfied, and smaller values indicate larger deviations.

**Monte Carlo-based Algorithm.** To compute precise point coordinates in geometric diagrams, we adopt a Monte Carlo-guided coordinate solver, which iteratively perturbs the coordinates of free points, propagates updates to dependent points via constraint functions, and minimizes geometric constraint violations. This approach effectively balances local optimization and global search, thereby avoiding entrapment in suboptimal configurations.

Formally, we define a geometric diagram as $S = (P_f, P_d, \mathcal{C})$. $P_f = \{p_1, \ldots, p_m\}$ denotes the set of free points whose coordinates $\{(x_1, y_1), \ldots, (x_m, y_m)\}$ can be independently adjusted. $P_d = \{p_{m+1}, \ldots, p_n\}$ denotes the set of dependent points, which are computed from the geometric constraints and the positions of $P_f$ via a constraint-solving function $\Phi$, i.e., $P_d = \Phi(P_f, \mathcal{C})$, where $\Phi$ encodes geometric relations such as parallelism, perpendicularity, incidence, etc.

For evaluation, we collect all scalar constraint scores into a single set

$$\mathcal{C} = C_{\text{lin\_rat}} \cup C_{\text{lin\_rel}} \cup C_{\text{ang\_val}} \cup C_{\text{ang\_rat}} \cup C_{\text{ang\_rel}}, \tag{6}$$

where each element $C_k \in \mathcal{C}$ is the value of one constraint metric defined in the five families above. By design, each $C_k$ is a normalized satisfaction score in the range $C_k \in [0, 1]$, and $C_k = 1$ indicates that the corresponding constraint is exactly satisfied.

The optimization objective is to maximize the overall satisfaction of all constraints. Given the set of constraint scores $\mathcal{C} = \{C_k\}$, we define the loss as

$$\mathcal{L}(S) = \max_{C_k \in \mathcal{C}} \left| 1 - C_k \right| \tag{7}$$

Since each $C_k \in [0, 1]$, minimizing $\mathcal{L}(S)$ is equivalent to maximizing the minimum constraint satisfaction score.

This loss measures how far the current diagram deviates from the target constraints. The algorithm performs inner-loop sampling ($Q$) and outer-loop global updates ($T$) on $P_f$ and $P_d$ until the deviation falls below the tolerance $\alpha$ (0.05). The full procedure is detailed in Appendix C.

**Geometric Diagram render.** To support high-quality visualization and compatibility with computational geometry tools, we use Matplotlib for rendering geometric diagrams. The renderer parses the formal language to extract points $P$ and line segments $L$, then applies a dynamic coordinate adjustment mechanism that computes appropriate scaling and translation based on the spatial distribution of points, which prevents overflow. Implementation details are provided in Appendix C.

## 4 EXPERIMENTS

**Implementation details.** The coordinate solver involves three principal hyperparameters: the loss convergence threshold $\alpha$, the number of inner loop iterations $Q$, and the number of outer loop iterations $T$. Specifically, we set $\alpha = 0.05$, $Q = 1000$, and $T = 1000$. Complete implementation details, including prompt formats and training configurations, are provided in the Appendix B.

### 4.1 RESULTS

We conduct a comprehensive evaluation of GeoLoom by benchmarking it against strong baselines. Additionally, we analyze the computational efficiency of GeoLoom, highlighting its scalability and inference speed.

**Qualitative evaluation.** Figure 4 presents representative visual examples comparing our method with two baselines: the state-of-the-art text-to-image model Seedream3.0 (Gao et al., 2025) and the domain-specific text-to-diagram generator AutomaTikZ (Belouadi et al., 2024). As shown, our approach, either training-free or fine-tuned, consistently generates diagrams that rigorously satisfy geometric constraints. In contrast, the baseline methods, though capable of producing diagram-like visuals, frequently violate these constraints. This superior performance arises from our integration of formal representations and a constraint-aware coordinate solver, ensuring both semantic alignment and geometric validity.

We further evaluate our method on more complex IMO-style datasets and compare with approaches that require human intervention, namely Penrose (Ye et al., 2020) and GeoGebra (Hohenwarter, 2007). As shown in Figure 5, all three methods can produce correct geometric diagrams; however, our method generates each diagram in about 60 seconds, whereas Penrose requires substantial additional time for users to learn and specify the problem in its complex symbolic language, and GeoGebra relies on manual construction, taking roughly 300 seconds per diagram.

**Quantitative evaluation.**

To complement the qualitative analysis, we quantified the error values between the generated diagrams and the target diagrams. We leverage our geometric constraint deviation score and group them into two metrics that can be used as quantitative indicators. The first is Line Consistency Index (LCI) capturing deviations related to line segments,

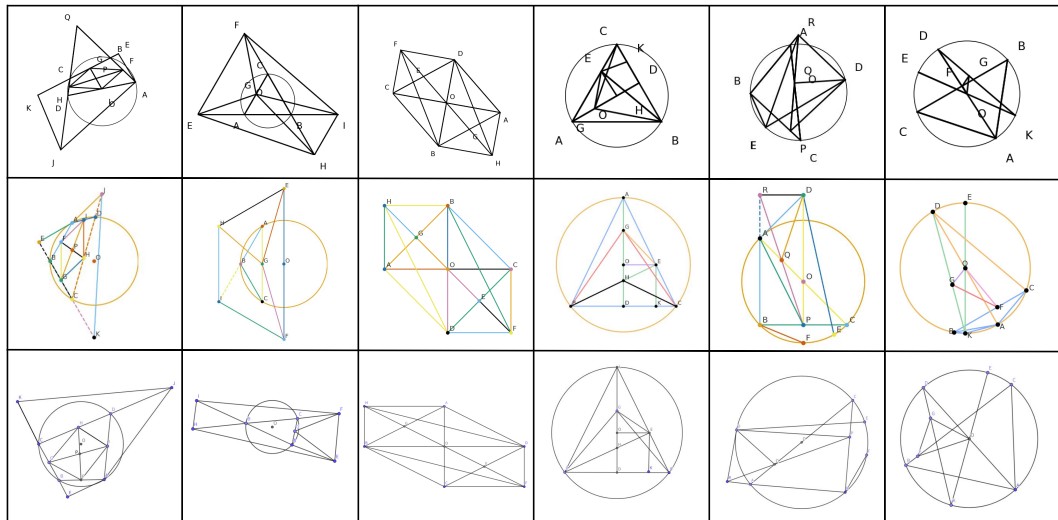

Figure 5: Complex examples and comparisons with Penrose and GeoGebra. GeoLoom, Penrose, and GeoGebra results arranged top to bottom. Natural language inputs appear in Appendix D.

Table 1: Quantitative comparison. True represents the accuracy of manual examination (Unit:%). Avg. represents the average value of LCI and ADI. The training-free model is DeepSeek-v3.

| Method | True | LCI | ADI | Avg. | Method | True | LCI | ADI | Avg. |
|---|---|---|---|---|---|---|---|---|---|
| *Fine-tuned* | | | | | *Fine-tuned* | | | | |
| LLaMa3.1-8b | 79.06 | 0.996 | 2.104 | 1.550 | Qwen2.5-7b | **85.34** | 0.747 | 1.818 | **1.283** |
| LLaMa3.2-3b | 77.21 | 0.951 | 2.104 | 1.549 | Qwen2.5-14b | 83.72 | 0.871 | 1.083 | 1.427 |
| LLaMa3-8b | 75.58 | 1.326 | 2.666 | 1.996 | *Training-free* | 81.16 | 0.926 | 1.747 | 1.337 |

$$LCI = 1 - \frac{1}{N_{line}} \left( \sum_{i=1}^{N_{lin\_rat}} \mathcal{C}_{lin\_rat}^i + \sum_{j=1}^{N_{lin\_rel}} \mathcal{C}_{lin\_rel}^j \right) \tag{8}$$

where $\mathcal{C}_{lin\_rat}$ and $\mathcal{C}_{lin\_rel}$ represent the length ratio and length relation constraint scores, respectively. $N_{line} = N_{lin\_rat} + N_{lin\_rel}$.

The second is Angle Deviation Index (ADI) capturing deviations related to angular properties,

$$ADI = 1 - \frac{1}{N_{angle}} \left( \sum_{k=1}^{N_{ang\_val}} C_{ang\_val}^k + \sum_{m=1}^{N_{ang\_rat}} C_{ang\_rat}^m + \sum_{n=1}^{N_{ang\_rel}} C_{ang\_rel}^n \right) \tag{9}$$

where $\mathcal{C}_{ang\_val}, \mathcal{C}_{ang\_rat}, \mathcal{C}_{ang\_rel}$ are represent angle value, angle ratio, and angle relationship constraints, respectively. The total number $N_{angle} = N_{ang\_val} + N_{ang\_rat} + N_{ang\_rel}$.

We conduct a comparative analysis of different LLMs under the finetuning setting and training-free, the results are shown in Table 1. To further validate the findings, we report human-evaluated generation accuracy, which supports the reliability of the proposed quantitative indicators. we organized a manual evaluation team to manually review the generated geometric diagrams. We use two methods: fine-tuning method and training free method to generate formal language, and use the formal language to generate geometric diagrams through a coordinate solver. The manual review results of the geometric diagrams generated by each method are shown in the Table 1. As shown in the table, the model finetuned with Qwen2.5-7b yields the highest performance, highlighting the advantage of incorporating formal language supervision during finetuning.

Table 3: Efficienty results (Unit:%). Unit of time is "s". Construct the training-free based on train set, average of the three fine-tuned Qwen2.5-7b on the test set, average of the three training-free based on the test set. (The time proportion of three experiments is shown in the Appendix D).

| Method | <10 | 11–20 | 21–30 | 31–40 | 41–50 | 51–60 | >60 |
|---|---|---|---|---|---|---|---|
| Training-free (train set) | 75.51 | 5.53 | 1.98 | 5.61 | 3.04 | 1.95 | 6.30 |
| Average (Training-free (test set)) | 72.13 | 6.71 | 2.65 | 1.71 | 5.38 | 4.45 | 6.94 |
| Average (Fine-tuned) | 71.36 | 7.85 | 2.83 | 2.83 | 1.94 | 1.46 | 11.73 |

**User study.** To evaluate the effectiveness from a human-centric perspective, we conduct a user study on a randomly selected sample of 100 geometric diagrams. Ten participants with backgrounds in mathematics or computer science were recruited to evaluate each diagram along two dimensions: (i) image quality, namely visual plausibility; and (ii) textual alignment, the consistency between the diagram and textual description. For each instance, participants were shown diagrams generated by four different methods (including ours) in the randomized order, accompanied by the original textual description. The fifth option, "None of the above," was provided to allow participants to reject all diagrams if none were satisfactory. Table 2 presents the aggregated results. Our method consistently outperformed all baselines across both evaluation criteria. A small proportion of samples were rated unsatisfactory by all participants, suggesting that evaluators exercised critical judgment rather than responding arbitrarily.

Table 2: User study in image quality and textual alignment. Quality represents the quality of the images. Alignment represents the textual alignment

| Model | Quality | Alignment |
|---|---|---|
| AutomaTikZ | 0 | 0 |
| Seedream | 0 | 0.2 |
| Ours (training-free) | 39.2 | 40.5 |
| Ours (fine-tuned) | 55.1 | 53.9 |
| None of the above | 5.7 | 5.4 |

**Efficiency results.** The Monte Carlo method greatly improves generation efficiency by enabling rapid sampling in complex solution spaces. To quantify this, we measured the per-instance generation time. As shown in Table 3, most diagrams were generated within 10 seconds, demonstrating strong practical efficiency. We further repeated the timing experiment on the test set in training-free and fine-tuned Qwen2.5-7b, and the resulting time distribution closely matched that of the train set in training-free, confirming stable and reliable performance across data settings.

## 4.2 ANALYSIS

**Failure case analysis.** We analyze failure cases of the Qwen2.5-7B experiments from both qualitative and quantitative perspectives. As shown in Figure 13, a sample is regarded as a failure if either the formalization or the generated diagram is incorrect. Most formalization errors arise from irregular or ambiguous natural-language descriptions that lead to topological mistakes. Even with correct formal inputs, the Monte Carlo optimization may still fall into local minima or produce overlapping configurations. Local minima constitute the primary failure mode, accounting for about 8% of the test set. More examples and statistics are provided in Appendix E.

**The effect of validation filters.** We here ablate the impact of validation filters on automatic formalization. As shown in Table 4, incorporating these filters into training-free autoformalization substantially improves the quality of the generated geometric diagrams, while markedly reducing both LCI and ADI scores. Notably, the filtered autoformalization results approach the quality of manually annotated formal languages. These findings demonstrate that validation filters effectively correct non-standard or noisy constraints in the autoformalization process, thereby significantly enhancing the overall reliability and expressiveness of the resulting formal language.

**The effect of different constraints.** To evaluate the impact of different geometric constraints on the final generation quality, we further conduct constraint ablation experiments. As shown in Table 4, retaining only length constraints yields the best length accuracy (lowest LCI) and the fastest convergence, indicating that length constraints are relatively robust and easy to satisfy. In contrast, when only angle constraints are preserved, the resulting angle quality is comparable to that under full constraints, suggesting that angle constraints are more sensitive and more difficult to optimize.

Table 4: Ablation study of filters and constraint. Labeled is a formal language for manual labeling.

| Formal language | Constraints | LCI↓ | ADI↓ | Average | Time |
|---|---|---|---|---|---|
| Training-free + Filter | Full(Length and Angle) | 0.962 | 1.747 | 1.337 | 13.379 |
| Training-free w/o Filter | Full | 1.340 | 1.864 | 1.602 | 15.945 |
| Labeled | Full | 0.870 | 1.537 | 1.203 | 15.782 |
| Labeled | Length( $C_{\text{lin\_rat}}$, $C_{\text{lin\_rel}}$ ) | 0.651 | - | - | 2.844 |
| Labeled | Angle( $C_{\text{ang\_val}}$, $C_{\text{ang\_rat}}$, $C_{\text{ang\_rel}}$ ) | - | 1.577 | - | 7.477 |

| Natural Language & Difference of formal language | Training-free | Fine-tuned | | | | |
|---|---|---|---|---|---|---|
| | | LLaMa3.1-8b | LLaMa3.2-3b | LLaMa3-8b | Qwen2.5-7b | Qwen2.5-14b |
| In rhombus ABCD, connect AC and BD, with diagonal lines AC and BD inter-secting at point O, AC=4, BD=8, point E on the edge Of AD, **3AE=AD**. Connect BE, BE intersects AC at point M. (I) **3AE=AD :** *LengthRatio(AE,AD,Ratio(1,3))* (II)**2AE=ED:** *LengthRatio(AE,ED,Ratio(1,2))* | | | | | | |
| | (I) | (I) | (I) | (I) | (II) | (I) |

Figure 6: Randomness in autoformalization and coordinate. Highlight the specific formal language that is different in fine-tuned and training-free methods. (More results are included in Appendix E).

**Randomness in autoformalization and coordinate solver.** Natural-language autoformalization is inherently ambiguous, so different but semantically equivalent formal expressions may be produced for the same input. These variations stem from the probabilistic nature of language models, yet all generated formalizations preserve the intended geometric semantics and constraints, enabling the downstream system to produce valid diagrams. This demonstrates the robustness of our autoformal-ization pipeline. Stochasticity also arises in the coordinate-solving stage: the Monte Carlo–based solver generates slightly different coordinates across runs. However, these variations remain within strict tolerance thresholds that ensure geometric correctness. Figure 6 illustrates these two controlled sources of variation and shows example diagrams generated under random sampling.

**Visualization of optimization process.** To illustrate the iterative convergence behavior of the Monte Carlo algorithm, we visualize intermediate states across iterations as sequential snapshots, forming a state-transition diagram (results provided in Appendix E). This representation offers an intuitive view of how the diagram progressively evolves toward the target configuration. To complement the structural visualization, Figure 7 presents the loss values at each

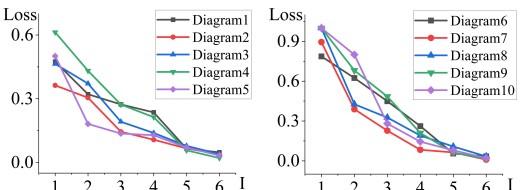

Figure 7: Loss curve during optimization.(The corresponding natural language and geometric diagrams can be found in Appendix E).

iteration, characterizing the optimization dynamics. The loss curve shows a steady downward trend, matching the improvement in geometric fidelity, indicating that the objective function effectively guides the search toward coherent solutions.

**Parameter analysis.** We investigate the impact of three key parameters, loss convergence thresh-old ($\alpha = 0.05$), inner-loop iterations ($Q = 1000$), and outer-loop iterations ($T = 1000$), on the efficiency and quality of geometric diagram generation. As shown in Figure 8, stricter convergence (smaller $\alpha$) yields higher accuracy but increases runtime. Larger $T$ enhances global exploration, while larger $Q$ improves local refinement; both improve diagram quality but incur additional com-putation. Overall, the results highlight a clear trade-off between optimization cost and geometric accuracy.

## 5 RELATED WORK

**Text-to-Image Generation.** Text-to-image generation has advanced rapidly with GANs (Zhang et al., 2017; Kang et al., 2023), autoregressive models (Chang et al., 2023; Ding et al., 2021), and diffusion approaches (Lian et al., 2024; Sun et al., 2024), achieving success in realistic image syn-thesis. However, these methods struggle with geometric diagrams that demand precise structure.

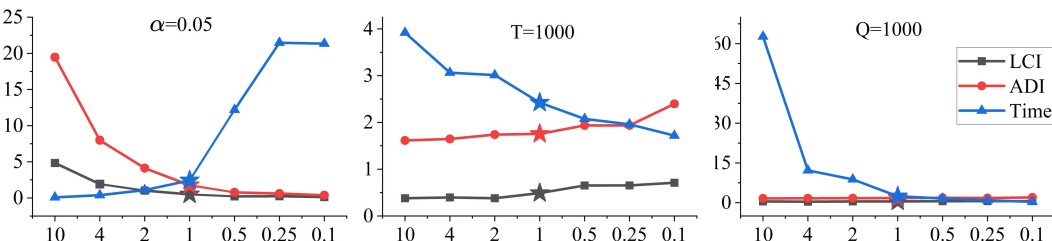

Figure 8: Parameter analysis.(x-axis: parameter scaling factor, y-axis: LCI, ADI, and time.

Existing diagram generation falls into layout-guided and code-guided paradigms. Layout-guided methods like DiagrammerGPT (Zala et al., 2023) use LLMs to plan layouts followed by layout-conditioned diffusion rendering. Code-guided methods generate domain-specific code (e.g., TikZ) from text, as in AutomaTikZ (Belouadi et al., 2024), DiagramAgent (Wei et al., 2025), and Magic-Geo (Wang et al., 2025), which enable programmatic diagram synthesis and editing. Penrose (Ye et al., 2020) provides formal-language-based automatic diagram generation, but it requires users to manually specify complex symbolic descriptions. GeoGebra (Hohenwarter, 2007) and Sketchpad (Carter, 2001), on the other hand, are typical dynamic geometry systems that depend on manual construction. Domain tools such as Mermaid and PlantUML support formal diagram construction but lack natural language understanding and geometric reasoning. Due to the sparsity of geometric diagrams, it is difficult for models and networks to accurately determine the relationships between geometric elements. In contrast, we propose GeoLingua, a generative-oriented formal language encoding geometric primitives, constraints, and dependencies. Integrated with a Monte Carlo-based coordinate solver, we propose GeoLoom, enabling high-fidelity diagram synthesis.

**Vector Graphics Generation.** Foundational work in vector graphics synthesis includes DeepSVG (Carlier et al., 2020), which introduced a hierarchical generative model to disentangle high-level shapes from low-level Bézier and line commands. Building on this, SVGFusion (Xing et al., 2024) proposed a scalable text-to-SVG framework that combines a vector-pixel fusion VAE with a diffusion transformer to learn a continuous latent space aligned with textual prompts. Recent advances such as Chat2SVG (Wu et al., 2025) integrate large language models and diffusion-based latent optimization in a two-stage pipeline, enabling enhanced visual fidelity and natural language–driven editing. NeuralSVG (Polaczek et al., 2025) further introduces an implicit neural representation of vector scenes via MLP encoding and score distillation, supporting inference-time control over layout and style with compositional semantics. Despite these advancements in aesthetics, editability (Lopes et al., 2019; Carlier et al., 2020; Cai et al., 2023; Wu et al., 2023), and scalability, most existing approaches lack explicit modeling of geometric constraints, such as parallelism, perpendicularity, and precise angle enforcement, limiting their applicability in domains requiring strict geometric correctness.

## 6 CONCLUSION

In this work, we address the longstanding challenge of automated geometric diagram generation by introducing GeoLoom. a novel framework that bridges natural language and precise geometric rendering through formal language-based reasoning. By leveraging GeoLingua, a generation-oriented formal language tailored for geometric domains, and constructing GeoNF, a high-quality paired dataset of natural language and formal representations, we establish a principled foundation for training and evaluation. GeoLoom's two-stage architecture, comprising a structured autoformalization module and a Monte Carlo-based coordinate solver, demonstrates strong spatial accuracy and computational efficiency. Through extensive experiments, we show that GeoLoom significantly outperforms existing baselines, making it a promising solution for scalable, accurate diagram generation in educational contexts. This work paves the way for future research in symbolic-geometric modeling and the integration of structured formalism into multimodal generation tasks.

Limitations remain: the coordinate solver is restricted to 2D Euclidean geometry, linguistic variability can still affect robustness, and an $8\%$ local-minimum failure rate persists. Addressing these issues will be important for future work.

## ETHICS STATEMENT

All work in this study strictly complies with the ICLR Code of Ethics. Comparative experiments were conducted using publicly available baseline methods, ensuring compliance with their licensing agreements. No personal or sensitive data were collected or used, and the study does not involve human subjects, privacy risks, or security concerns. The potential societal impact of this research was carefully evaluated, confirming conformity with academic ethical standards. All authors declare no conflicts of interest. We affirm our commitment to academic integrity, and the results presented are original with all relevant references properly cited.

## REPRODUCIBILITY STATEMENT

We have taken concrete measures to ensure reproducibility. Section 3 details the full method architecture and implementation objectives, while Section 4 provides the experimental setup and parameter configurations. Additional resources are included in the appendix, comprising the formal language definitions, pseudocode for the Monte Carlo–based coordinate solver, prompt engineering strategies used in autoformalization, and the implementation of validation filters. Together, these materials are intended to enable readers to independently reproduce our results.

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

Table 5: Formal language definitions for "shapes" of GeoLingua.

| Natural Language | Formal Language |
|---|---|
| Circle /⊙, radius=r | Circle(center_name,radius_value) |
| Triangle | Polygon(triangle_name) |
| Parallelogram | Parallelogram(parallelogram_name) |
| Rhombus | Rhombus(rhombus_name) |
| Square | Square(square_name) |
| Rectangle | Rectangle(rectangle_name) |
| Trapezoid | Polygon(trapezoid_name) |
| Sector | Sector(sector_name,the angle of sector,radius_value) |
| String of circle | StringOfCircle(circle_center,string_name) |
| Circle inscribed polygon | InscribedPolygon(circle_center,polygon_name) |
| The inscribed circle of a polygon | CircumscribedPolygon(circle_center,polygon_name) |
| Other polygon | Polygon(polygon_name) |

Table 6: Formal language definitions for "depengence" of GeoLingua.

| Natural Language | Formal Language |
|---|---|
| Point at line segment | PointAtLine(point_name,line_name,0) |
| Proportional point of line segment | PointAtLine(point_name,line_name,ratio_value) |
| Intersection | LineIntersect(line_name1,line_name2,intersection) |
| Point on the arc | PointAtArc(point_name,arc_name,arc_type,) |
| Point P on the circle | PointAtArc(point_name,circle_center,0) |
| Extend the points on the line | ExtensionLine(extension-line_name,point_name) |
| Tangent of circle | Tangent(circle_center,point of tangency,tangent) |
| Make perpendicular line at point | DoPerpendicular(point_name,line_name,foot) |
| Point inside polygon | PointInShape(point_namw,polygon_name,0) |

## A  FORMAL LANGUAGE FRAMEWORK

This chapter will provide a detailed introduction to the framework for defining formal languages (Section 2).

The formal language statements of the four module in GeoLingua are shown in the Table 5, Table 6, Table 7 and Table 8. Geometric names, length values, and angle values in formal language can be adjusted according to actual situations.

## B  TRAINING-FREE DETAILS

This section will provide a detailed introduction to prompt formats and validation-based filter of the training-free (Section 3.1).

### B.1  PROMPT FORMATS

For the design of prompt, we have divided it into four parts: role, formal language framework, examples, and requirements. The content of each section is as follows:

- **Role:** You are a professional formal language translation assistant capable of accurately converting natural language descriptions (natural language) of geometry problems into structured formal language (formal language). The conversion rules from natural language to formal language are as follows:

- **Formal Language Framework:** A formal language framework in the form of markdown.

- **Examples:** Listed the diverse geometric natural language and formal language.

- **Requirements:** You must strictly follow the conversion rules defined above and special conversion rules for the conversion. The final output JSON format is as follows : "JSON data format". The natural language description entered by the user is as follows:" Among

Table 7: Formal language definitions for "length constraint" of GeoLingua.

| Natural Language | Formal Language |
|---|---|
| Segment length | Length(line_name,length_value) |
| Line segment ratio | LengthRatio(line_name1,line_name2,Ratio(value1,value2)) |
| Arc ratio | ArcRatio(arc_name1,arc_name2,Ratio(value1,value2)) |
| Line segment relationship | LengthAddandSub((left side ),relation,(right side)) |
| Perimeter of polygon | LengthAddandSub((lines_name of polygon),equal,perimeter) |
| Distance from point to line | PointLineDistance(point_name,line_name,distance_value) |
| Connect line segment | ConnectPoints(line_name) |

Table 8: Formal language definitions for "angle constraint" of GeoLingua.

| Natural Language | Formal Language |
|---|---|
| $\parallel$ | Parallel(line_name1,line_name2,0) |
| $\perp$ | Perpendicular(line_name1,line_name2,90) |
| Degree of angle | Angle(angle_name,degree_value) |
| Trigonometric function | TriFunction(function,angle_name,value) |
| Angle ratio | AngleRatio(angle_name1,angle_name2,Ratio(value1,value2)) |
| Angle relationship | AngleAddandSub((left side),relation,(right side)) |

them, the JSON data format is: "shapes": [], "dependence": [], "length constraint": [], "angle constraint": []

## B.2 VALIDATION-BASED FILTER

There is an illusion problem with large models. In order to prevent the model from outputting undefined formal statements, we set validation-based filter to ensure that the formal language content meets the defined requirements. Validation-based filters are mainly designed to prevent tampering or the addition of formal language statements to the Large Language model. We have set up filters three times. If it does not meet the requirements of formal language, the prompt will be reloaded until all three chances are exhausted. If required, the formal language will be outputted. The implementation technology framework is shown as Figure 9.

## C COORDINATE SOLVER

### C.1 MONTE CARLO-BASED ALGORITHM

In this section, We have provided a detailed introduction to the implementation algorithm of a coordinate solver driven by Monte Carlo (Monte Carlo-based Algorithm of the Section 3.2). The specific algorithm process is shown in Algorithm 1.

### C.2 GEOMETRIC DIAGRAM RENDER ALGORITHM

This section will provide a detailed introduction to geometric rendering algorithms(Geometric Diagram render of the Section 3.2).

We use the regular matching method to extract the points identifiers $P$ and all line segment representations $L$ in the geometric diagram from the formal language. By solving the coordinates, we obtained the coordinates $P = \{p_1(x_1,y_1),\ldots,p_m(x_m,y_m)\}$ of each point. In order to better present the geometric diagram, we designed an adaptive canvas. The adaptive canvas algorithm is as Algorithm 2

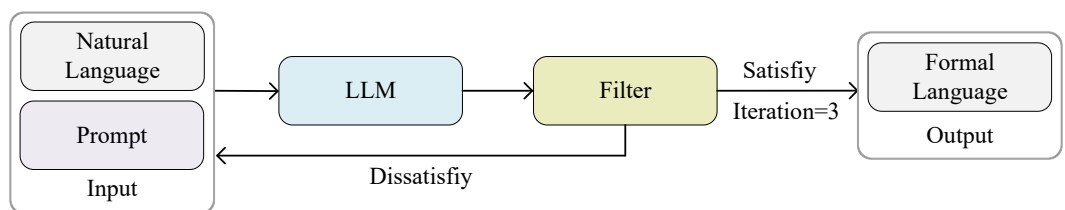

Figure 9: Technical framework of the validation-based filter.

---

**Algorithm 1:** Monte Carlo-based algorithm

---

**Input:** Textual description $T$
**Output:** Geometric diagram $S_{global}$

1 Initialize geometric diagram $S^0 = (P_f^0, P_d^0, \mathcal{C})$ and Calculate the initial loss value $\mathcal{L}(S^0)$;

2 Define the global optimal solution as the initial state $S_{global} = S^0$;

3 **while** $t < T \text{ or } \mathcal{L}(S)_{max} > \alpha$ **do**

4     for the (t-1)-th iteration result: $S^{t-1} = (P_f^{t-1}, P_d^{t-1}, \mathcal{C})$ ;

5     **while** $q < Q$ **do**

6         $S^q = S^{t-1}$;

7         $\mathcal{L}(S)_{best} = \mathcal{L}(S^q)_{max}$;

8         Monte Carlo perturbation free point $P_f^q$;

9         $\mathcal{L}(S^q)_{max} = \text{evaluate}(S^q, P_f^q, P_d^q)$;

10         **if** $\mathcal{L}(S^q)_{max} < \mathcal{L}(S)_{best}$ **then**

11             Update local optimal solution: $\mathcal{L}(S)_{best} = \mathcal{L}(S^q)_{max}$;

12             $S_{local} = S^q$ ;

13         **end**

14     **end**

15     Update the dependency points based on the local optimal solution: $S^{(t)} = S_{local}$;
    $P_d^{(t)} = \Phi(P_f^t, \mathcal{C})$;

16     $\mathcal{L}(S^t)_{max} = \text{evaluate}(S^t, P_f^t, P_d^t)$;

17     **if** $\mathcal{L}(S^t)_{max} < \mathcal{L}(S)_{best}$ **then**

18         Update the global optimal solution: $\mathcal{L}(S)_{best} = \mathcal{L}(S^t)_{max}$;

19         $S_{global} = S^t$ ;

20     **end**

21     **return** $S_{global}$;

22 **end**

---

## D EXPERIMENT RESULT

In this section, we will briefly introduce the formal language corresponding to geometric diagrams in qualitative evaluation, baseline methods, and efficiency results in quantitative evaluation (Section 4.1).

### D.1 QUALITATIVE EVALUATION EXPERIMENT

The additional results in the qualitative evaluation experiment are shown in Figure 10. (The other resultds of qualitative evaluation is shown in the Figure 4 of Section 4.1). The formal languages corresponding to the geometric diagrams are shown as Figure 11. The natural-language descriptions corresponding to Figure 5 in Section 4.1 are shown in the Figure 12.

### D.2 BASELINE METHODS:

In the comparative experiment, we have compared our method with the following baselines:

1) **AutomaTiKZ:** AutomaTiKZ is an abstract graphics language based on vector drawing, which describes geometric shapes, charts, scientific diagrams, etc. through code, compiles

---

**Algorithm 2:** Geometric Diagram Render Algorithm

**Input:** Geometric diagram $S_{global}$.
**Output:** Image of geometric diagram.

1 extract all points coordinate $P = \{p_1(x_1, y_1), \ldots, p_m(x_m, y_m)\}$ from $S_{global}$.
2 Regularly parsing points $P$ and lines $L$ from formal languages.
3 **step1: Calculate coordinate offsets**.
4 $x_f$ = -min(x) if $\min(x) < 0$ else 0;
5 $y_f$ = -min(y) if $\min(y) < 0$ else 0;
6 **step2: Apply coordinate translation**.
7 **if** $x_f \neq 0$ *or* $y_f \neq 0$ **then**
8     **for** *each point* $p \in P$ **do**
9         $x \leftarrow +x_f$;
10         $y \leftarrow y + y_f$;
11     **end**
12     Update coordinate sets: $P = \{p_1(x_1, y_1), \ldots, p_m(x_m, y_m)\}$;
13 **end**
14 **step3: Determine canvas size** $C_{size}$.
15 $M_{coord} \leftarrow \max(\max(x), \max(y))$;
16 **if** $M_{coord} > C_{size}$ **then**
17     $C_{size} \leftarrow (\lfloor M_{coord}/50 \rfloor + 1) \times 50$;
18     50 is used to ensure that the canvas is an integer multiple of 50
19 **end**
20 Draw a geometric diagram with lines $L$ and adjusted coordinates $P$ on a new canvas $C_{size}$.
21 **return** Geometric diagram;

---

| Natural Language | Ours (training-free) | Ours (fine-tuned) | AutomaTikZ | Seedream |
|---|---|---|---|---|
| In diamond shapes ABCD, AC and BD intersect at points O, E, and F, which are the midpoints of OA and OC, respectively, Connect DF and BE | | | | |
| In square ABCD, the side length of square ABCD is 4, connecting AC and BD at point O, extending DC to point E, connecting AE, ∠CAE=15° | | | | |

Figure 10: Additional result of quantitative evaluation. The baseline methon are AutomaTikz (base LLaMa-7b) and SeeDream model.

and outputs scalable vector graphics. We choose the TiKZ-LLaMa7b model trained on LLaMa as base model.

2) **SeeDream:** Basic model for bilingual image generation in Chinese and English

3) **Penrose:** Penrose describes geometric and mathematical relationships using the Substance language, and then automatically generates a diagram that satisfies these constraints through optimization. However, its drawback is that users must first learn and manually specify this relatively complex description language.

4) **GeoGebra:** GeoGebra relies on manual diagram construction, which can ensure high visual quality but requires substantial human effort and time.

The AutomaTiKZ and SeeDream methods perform fully automatic geometric diagram generation, Penrose and GeoGebra require human intervention. On standard benchmark datasets(GeoNF), we compare our approach with AutomaTiKZ and SeeDream and show that it can generate correct diagrams both rapidly and reliably. On more challenging datasets such as IMO-style problems, we compare our method with Penrose and GeoGebra. Although these baselines are able to produce high-quality diagrams, under comparable visual quality our method is faster and does not require any manual involvement.

| Natural lanuage | Formal language | | | |
|---|---|---|---|---|
| | shapes | dependence | length constraint | angle constraint |
| The triangle ABC is an inscribed equilateral triangle of circle O, point O is the center, points D and E are on sides AC and AB respectively, connecting OD and OE, DA=BE. | "Circle(O,r)", "InscribedPolygon(O,ABC)" | "PointAtLine(D,AC,0)", "PointAtLine(E,AB,0)"], | "ConnectPoints(OD)", "ConnectPoints(OE)", "LengthRatio(AB,BC,Ratio(1,1))", "LengthRatio(AB,AC,Ratio(1,1))", "LengthRatio(DA,EB,Ratio(1,1))" | [] |
| In triangle ABC, points D, E, F are respectively the midpoints of AB, AC, and BC, Connecting DE, EF, and DF. Points P, M, and N are respectively the midpoints of DE, DF, and EF, connecting PM, PN, and MN. | "Polygon(ABC)" | "PointAtLine(D,AB,2)", "PointAtLine(E,AC,2)", "PointAtLine(F,BC,2)", "PointAtLine(P,DE,2)", "PointAtLine(M,DF,2)", "PointAtLine(N,EF,2)"], | "ConnectPoints(DE)", "ConnectPoints(EF)", "ConnectPoints(DF)", "ConnectPoints(PM)", "ConnectPoints(PN)", "ConnectPoints(MN)" | [] |
| In the four sides ABCD, ∠BAD=120°, ∠ABC=∠ADC =90°, find points M and N on BC and CD respectively, and connect AM and AN. | "Polygon(ABCD)" | "PointAtLine(M,BC,0)", "PointAtLine(N,CD,0)" | "ConnectPoints(AM)", "ConnectPoints(AN)", | "Angle(BAD,120)", "Perpendicular(BA,BC,90)", "Perpendicu-lar(DA,DC,90)" |
| I n rectangle ABCD, point E is on edge AB, connected to DE, D E is the bisector of ∠ADC. Point F is on the extension of DE, connected to BF, ∠BFE = 90°, connected to AF, CF, CF, and AB intersecting with G. | "Rectangle(ABCD) | "PointAtLine(E,AB,0)", "ExtensionLine(DE,F)", "LineIntersect(CF,AB,G)" | "LengthRatio(AD,BC,Ratio(1,1))", "LengthRatio(AB,DC,Ratio(1,1))", "ConnectPoints(DE)", "ConnectPoints(BF)", "ConnectPoints(AF)", "ConnectPoints(CF)" | "Perpendicular(AB,AD,90)", "Perpendicular(DA,DC,90)", "Perpendicular(FB,FE,90)", "AngleRatio(ADE,CDE, Ratio(1,1))" |
| I n diamond shapes ABCD, AC and BD intersect at points O, E, and F, which are the midpoints of OA and OC, respectively, c onnect DF and BE | "Rhombus(ABCD)" | "LineIntersect(AC,BD,O)", "PointAtLine(E,OA,2)", "PointAtLine(F,OC,2)" | "LengthRatio(AB,CD,Ratio(1,1))", "LengthRatio(BC,AD,Ratio(1,1))", "LengthRatio(AB,BC,Ratio(1,1))", "ConnectPoints(AC)", "ConnectPoints(BD)", "ConnectPoints(DF)", "ConnectPoints(BE)" | "Parallel(AB,DC,0)", "Parallel(AD,BC,0)" |
| In square ABCD, the side length of square ABCD is 4, connecting AC and BD at point O, extending DC to point E, connecting AE, ∠CAE=15° | "Square(ABCD)" | "LineIntersect(AC,BD,O)", "ExtensionLine(DC,E)" | "LengthRatio(AB,AD,Ratio(1,1)), LengthRatio(AB,BC,Ratio(1,1))", "Length(AB,4)", "ConnectPoints(AE)" | "Perpendicular(AB,AD,90)", "Perpendicular(DA,DC,90)", "Angle(CAE,15)" |

Figure 11: The formal language corresponding to natural language in Figure4 and Figure10.

Table 9: Efficiency result of three time proportion(Unit:%). Unit of time is "s". Three experiment of fine-tuned Qwen2.5-7b on the test set, three training-free experiments based on the test set.

| Method | The number of Experiments | <10 | 11–20 | 21–30 | 31–40 | 41–50 | 51–60 | >60 |
|---|---|---|---|---|---|---|---|---|
| Training-free (test) | the first experiment | 70.49 | 6.32 | 3.28 | 2.34 | 6.79 | 3.51 | 7.26 |
| | the second experiment | 71.90 | 7.26 | 1.87 | 1.64 | 4.22 | 5.39 | 7.73 |
| | the third experiment | 74.00 | 6.56 | 2.81 | 1.17 | 5.15 | 4.45 | 5.85 |
| Fine-tuned | the first experiment | 71.36 | 8.25 | 3.40 | 0.97 | 2.18 | 1.94 | 11.89 |
| | the second experiment | 71.84 | 7.52 | 2.67 | 4.61 | 2.91 | 1.46 | 8.98 |
| | the third experiment | 70.87 | 7.77 | 2.43 | 2.91 | 0.73 | 0.97 | 14.32 |

## D.3 EFFICIENCY RESULTS

In the efficiency experiment (Efficiency result of the Section 4.1), three times experiments were conducted on the fine-tuned Qwen2.5-7b model and the training-free DeepSeek-V3, and the time consumption of the three times experiments is shown in Table 9 (The average time consumption of the three experiments is shown in Table 3 of Section 4.1). The three experiments in Table 9 indicate that the majority of geometric graph generation takes less than 10 seconds. The results of the time proportions of the two methods indicate that the efficiency of the coordinate parser is highly stable.

## E EXPERIMENT ANALYSIS

In this section, We will supplement necessary experimental analysis (Section 4.2).

### E.1 FAILURE CASE OF GEOLOOM

- Topological of formal language: For a very small number of natural-language descriptions that are not written in a standard form, the automatic formalization process can produce errors in the topological ordering. As shown by the example on the right in Figure 13, the natural-language description uses DE ∥ AB, DF ∥ AC, while the standardized description should be DE ∥ BA, DF ∥ CA. Since the topological order is not corrected during formal-

| Natural lanuage | Formal language | | | |
|---|---|---|---|---|
| | shapes | dependence | length constraint | angle constraint |
| Let ABCD be a cyclic quadrilateral with circumcircle O, angle ABC = 120°, and angle CDA = 60°. Extend CD beyond D to a point J such that DJ = AD. Extend BC beyond C to a point K such that CK = BC. Join AJ and KJ. Extend CB beyond B to a point E and join AE, such that angle AEB = 90°. Let F, G, H, I be the midpoints of AB, BC, CD, DA, respectively. Join FG, GH, HI, IF, FH, and GI. Let FH and GI intersect at P, and let BA and DC intersect at Q. Prove that FGHI is a parallelogram. | "Circle(O,r)", "InscribedPolygon(O,ABCD)" | "ExtensionLine(BC,K)", "ExtensionLine(CD,J)", "ExtensionLine(CB,E)", "PointAtLine(F,AB,2)", "PointAtLine(G,BC,2)", "PointAtLine(H,CD,2)", "PointAtLine(I,DA,2)", "LineIntersect(FH,GI,P)", "LineIntersect(AB,DC,Q)" | "LengthRatio(DJ,AD,Ratio(1,1))", "LengthRatio(CK,BC,Ratio(1,1))", "ConnectPoints(AE)", "ConnectPoints(FG)", "ConnectPoints(GH)", "ConnectPoints(HI)", "ConnectPoints(IF)", "ConnectPoints(AJ)", "ConnectPoints(KJ)" | "Angle(ABC,120)", "Angle(ADC,60)", "Angle(AEB,90)" |
| Let ABC be a triangle inscribed in circle O, with AB = BC. Point F lies on line BC such that CF = BC. Point E lies on ray BA beyond A such that BA = AE. Point G is the midpoint of AC. Join EF, EG, and GF. Extend CB to a point H, and extend AB to a point I. Join HE, HG, HI, and FI. The construction is such that HI is parallel to EF. Prove that quadrilateral HIEF is an isosceles trapezoid. | "Circle(O,r)", "InscribedPolygon(O,ABC)" | "ExtensionLine(BC,F)", "ExtensionLine(BA,E)", "ExtensionLine(CB,H)", "ExtensionLine(AB,I)", "PointAtLine(G,AC,2)" | "LengthRatio(AB,BC,Ratio(1,1))", "LengthRatio(CF,BC,Ratio(1,1))", "LengthRatio(AE,BA,Ratio(1,1))", "LengthRatio(BH,BI,Ratio(1,1))", "ConnectPoints(EG)", "ConnectPoints(FG)", "ConnectPoints(HG)", "ConnectPoints(IG)", "ConnectPoints(EF)", "ConnectPoints(HE)", "ConnectPoints(HI)", "ConnectPoints(IF)" | "Parallel(HI,EF,0)" |
| In a rhombus ABCD, the diagonals AC and BD intersect at O. Let E be the midpoint of CD, and G the midpoint of AB. Extend segment OE beyond E to a point F such that OE = EF. Extend segment OG beyond G to a point H such that OG = GH. Prove that HDBF is a parallelogram, and that quadrilaterals AHBO and DOCF are squares. | "Rhombus(ABCD)" | "LineIntersect(AC,BD,O)", "PointAtLine(E,CD,2)", "PointAtLine(G,AB,2)", "ExtensionLine(OE,F)", "ExtensionLine(OG,H)" | "LengthRatio(AB,CD,Ratio(1,1))", "LengthRatio(BC,AD,Ratio(1,1))", "LengthRatio(AB,BC,Ratio(1,1))", "LengthRatio(OE,EF,Ratio(1,1))", "LengthRatio(OG,GH,Ratio(1,1))", "ConnectPoints(DF)", "ConnectPoints(CF)", "ConnectPoints(AH)", "ConnectPoints(BH)", "ConnectPoints(DH)", "ConnectPoints(BF)" | "Parallel(AB,DC,0)", "Parallel(AD,BC,0)", "Angle(ABC,100)" |
| In circle O, triangle ABC is inscribed and satisfies AB = AC. Through B, draw BE perpendicular to AC with foot E on AC. Through A, draw AD perpendicular to BC with foot D on BC. Through E, draw EK perpendicular to BC with foot K on BC. Join O and E; suppose OE is perpendicular to AD. Lines BE and AD intersect at H, and it is given that AH = 2·OD. Point G lies on segment AD. Join BG, CG, and GE. | "Circle(O,r)", "InscribedPolygon(O,ABC)" | "PointAtLine(E,AC,0)", "PointAtLine(D,BC,0)", "PointAtLine(K,DC,0)", "PointAtLine(G,AD,0)", "LineIntersect(BE,AD,H)" | "LengthRatio(AB,AC,Ratio(1,1))", "LengthRatio(AH,OD,Ratio(2,1))", "ConnectPoints(BG)", "ConnectPoints(EG)", "ConnectPoints(EO)", "ConnectPoints(GC)" | "Perpendicular(AD,BC,90)", "Perpendicular(BE,AC,90)", "Perpendicular(EO,AD,90)", "Perpendicular(EK,BC,90)" |
| In circle O, an equilateral triangle ABC is inscribed in circle O, and angle ABC = 90°. Chord DE lies on circle O, and point P is a point on segment BC. Extend BA beyond A to a point R, and join PR. Let PR intersect AC at Q. Join BE, RE, DR, and DQ, with DR perpendicular to AC. Join DP, and let DP be perpendicular to BC. | "Circle(O,r)", "InscribedPolygon(O,ABC)", "StringOfCircle(O,DE)" | "PointAtLine(P,BC,0)", "ExtensionLine(BA,R)", "LineIntersect(RP,AC,Q)" | "PointLineDistance(O,DE,0)", "ConnectPoints(DR)", "ConnectPoints(RE)", "ConnectPoints(RP)", "ConnectPoints(DQ)", "ConnectPoints(DP)" | "Angle(ABC,90)", "Perpendicular(DR,AB,90)", "Perpendicular(DQ,AC,90)", "Perpendicular(DP,BC,90)" |
| In circle O, triangle ABC is inscribed with angle BAC = 80°. KE is a diameter of circle O. Another diameter AD bisects angle BAC. Point F is the midpoint of BC, and point G is the midpoint of segment DK. Join DK, FG, GO, and FO. It is given that FG = FO. | "Circle(O,r)", "InscribedPolygon(O,ABC)", "StringOfCircle(O,KE)", "StringOfCircle(O,AD)" | "PointAtLine(F,BC,0)", "PointAtLine(G,DK,0)" | "PointLineDistance(O,KE,0)", "PointLineDistance(O,AD,0)", "LengthRatio(FG,FO,Ratio(1,1))", "ConnectPoints(FG)", "ConnectPoints(GO)", "ConnectPoints(FO)", "ConnectPoints(DK)" | "Angle(BAC,80)", "AngleRatio(BAD,CAD,Ratio(1,1))" |

Figure 12: The formal language corresponding to natural language in Figure5.

ization, the geometric diagram cannot be constructed correctly, and the evaluation metrics become extremely abnormal. Such cases account for 3% of the test set.

- Local minima in Monte Carlo coordinate optimization: Since the geometric constraint objective is inherently highly non-convex (especially when multiple angle, ratio, collinearity, and concyclicity constraints are involved), the optimization process is inevitably prone to local minima. The local minimum solution is characterized by evaluation metrics that are close but do not meet the threshold, and geometric diagrams that visually resemble the target image but do not meet the required details. In the middle example of Figure 13, the segment ratio constraint is correctly specified in the formal language. However, during random optimization, the target ratio of 2:1 cannot be reached, causing the procedure to converge to a local minimum.

- Overlapping problem: When two points are too close to each other, a quasi-overlapping issue arises. Although such cases still satisfy the quantitative evaluation metrics, they are visually unsatisfactory for practical use, as shown in Figure 13. These failure cases account for 3% of the test dataset.

## E.2 RANDOMNESS IN AUTOFORMALIZATION AND COORDINATE SOLVER

In the randomness in autoformalization and coordinate solver experimental (Randomness in autoformalization and coordinate solver of The Section 4.2), The supplementary experimental results are as figure 14 (The other results are shown in Figure 6 of Section4.2). The formal language corresponding to the geometric diagram is as figure15.

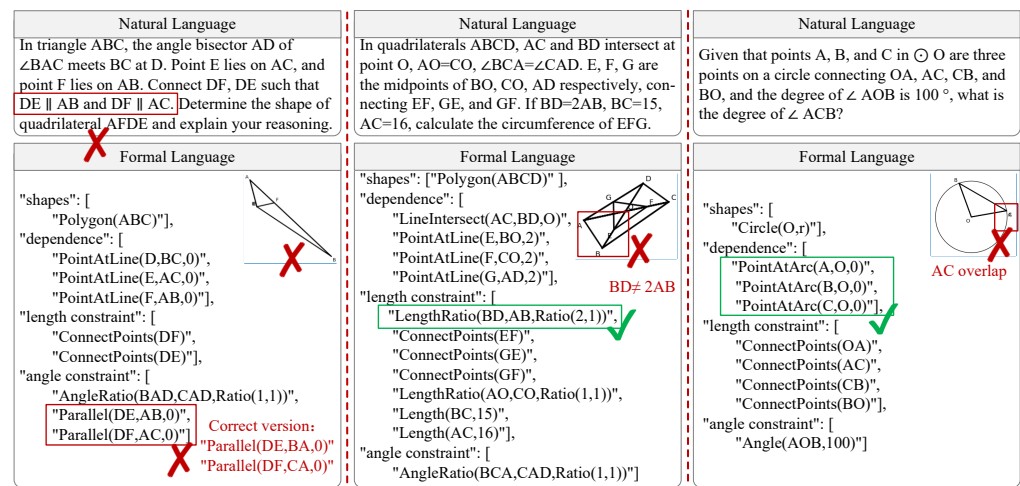

Figure 13: Analysis of Failure Cases. On the left is a case of topological of formal language. In the middle is a case of local minima in Monte Carlo coordinate optimization, and on the right is a case of overlapping problem.

| Natural Language & Different of formal language | Training-free | Fine-tuned | | | | |
|---|---|---|---|---|---|---|
| | | LLaMa3.1-8b | LLaMa3.2-3b | LLaMa3-8b | Qwen2.5-7b | Qwen2.5-14b |
| In rectangles ABCD, point E is on line segment CD, point F is on the extension of line segment AB, connecting EF to line segment BC at point G, and connecting BD, **DE=BF=2**. (I)**DE=BF=2:** *Length(DE,2), Length(BF,2)* (II)**DE=2,DE:BF=1:1:** *Length(DE,2), LengthRatio(DE,BF,Ratio(1,1))* | (I) | (II) | (I) | (II) | (II) | (I) |
| In triangle ABC, AB=AC, **make AD perpendicular to BC at point D**, If ∠BAC=70°. (I)**make AD⊥BC:** *DoPerpendicular(A,BC,D)* (II)**D at line BC, AD⊥BC:** *PointAtLine(D,BC,0), Perpendicular(AD,BC,90)"* | (I) | (II) | (I) | (I) | (II) | (I) |

Figure 14: Additional result of randomness in autoformalization and coordinate solver. Highlight formal languages and display geometric diagrams through fine-tuned and training-free.

The diversity of mathematical expressions for the same concept leads to the diversity of formal languages. The differences in the results of automformalization are detailed in Figure 14, and highlighted the differences. The experimental results indicate that autoformalization may achieve different descriptions but semantically equivalent formal languages, the formal languages with equal semantics, coordinate solver can result in geometric diagrams with text matching.

### E.3 VISUALIZATION OF OPTIMIZATION PROCESS

In the visualization of optimization process experiment (Visualization of optimization process of the Section 4.2), the geometric optimization process corresponding to the loss curve is shown in Figure 17. The Figure 17 lists the initialization state of the geometric diagram, the intermediate optimization snapshot, and the global optimal solution.The formal language corresponding to the geometric diagram is as Figure 16.

| Natural lanuage | Formal language | | | |
|---|---|---|---|---|
| | shapes | dependence | length constraint | angle constraint |
| In rhombus ABCD, connect AC and BD , with diagonal lines AC and BD intersecting at point O, AC=4, BD=8, point E on the edge Of AD, 3AE=AD. BE intersects AC at point M. | "Rhombus(ABCD)" | "LineIntersect(AC,BD,O)", "PointAtLine(E,AD,0)", "LineIntersect(BE,AC,M)" | "ConnectPoints(AC)", "ConnectPoints(BD)", "Length(AC,4)", "Length(BD,8)", "LengthRatio(AE,AD,Ratio(1,3))", "LengthRatio(AB,CD,Ratio(1,1))", "LengthRatio(BC,AD,Ratio(1,1))", "LengthRatio(AB,BC,Ratio(1,1))" | "Parallel(AB,DC,0)", "Parallel(AD,BC,0)" |
| In rectangles ABCD, point E is on line segment CD, point F is on the extension of line segment AB, connecting EF to line segment BC at point G, connecting BD, DE=BF=2 | "Rectangle(ABCD)" | "PointAtLine(E,CD,0)", "ExtensionLine(AB,F)", "LineIntersect(EF,BC,G)" | "ConnectPoints(EF)", "ConnectPoints(BD)", "Length(DE,2)", "Length(BF,2)" | [] |
| In triangle ABC, AB=AC, make AD perpendicular to BC at point D, ∠BAC=70 °. | "Polygon(ABC)" | "DoPerpendicular(A,BC,D)" | "LengthRatio(AB,AC,Ratio(1,1))" | "Angle(BAC,70)" |

Figure 15: The formal language corresponding to natural language in Figure6 and Figure14

| Natural lanuage | Formal language | | | |
|---|---|---|---|---|
| | shapes | dependence | length constraint | angle constraint |
| In △ ABC, ∠ ACB=90 °, ∠ ABC=30 °, AD bisect ∠BAC , A D passing hrough A intersects BC at D, DE ⊥ AB , perpendicular foot is E, DE=1. | "Polygon(ABC)" | "PointAtLine(D,AC,0)", "PointAtLine(E,AB,0)"], | "Length(AB,8)", "Length(BC,12)", "ConnectPoints(BD)", "ConnectPoints(AE)", "ConnectPoints(CE)" | "Perpendicular(BA,BC,90)", "AngleRatio(ABD,BCE, Ratio(1,1))" |
| In △ ABC, ∠ ABC=90 °, AB=8, BC=12, D is a moving point on the AC side, connecting BD. E is a moving point on BD, connecting AE and CE, ∠ ABD=∠ BCE. | "Polygon(ABC)" | "PointAtLine(D,BC,0)", "DoPerpendicular(D,AB,E)" | "Length(DE,1.5)", "Length(BD,3)" | "Perpendicular(CA,CB,90)", "AngleRatio(BAD,DAC, Ratio(1,1))" |
| In △ ABC, ∠ ACB=90 °, AD divides ∠ BAC equally and intersects BC with D, DE is perpendicular to AB with E. DE=1.5, BD=3. | "Polygon(ABC)" | "PointAtLine(D,BC,0)", "DoPerpendicular(D,AB,E)" | "Length(DE,1.5)", "Length(BD,3)" | "Perpendicular(CA,CB,90)", "AngleRatio(BAD,DAC, Ratio(1,1))" |
| In the parallelogram ABCD, BD is the diagonal, BD=CD, ∠ BCD=70 °, AE is perpendicular to BD at point E. | "Parallelogram(ABCD)" | "DoPerpendicular(A,BD,E)" | "LengthRatio(AB,CD,Ratio(1,1))", "LengthRatio(BC,AD,Ratio(1,1))", "LengthRatio(BD,CD,Ratio(1,1))" | "Parallel(AB,DC,0)", "Parallel(AD,BC,0)", "Angle(BCD,70)" |
| In parallelogram ABCD, the diagonal lines AC and BD intersect at point O, AB ⊥ AC. Point H is on the BD, AH ⊥BD, AB=2, BC=$2sqrt{3}$. | "Parallelogram(ABCD)" | "LineIntersect(AC,BD,O)", "PointAtLine(H,BD,0)", | "LengthRatio(AB,CD,Ratio(1,1))", "LengthRatio(BC,AD,Ratio(1,1))", "LengthRatio(BD,CD,Ratio(1,1))", "Length(AB,2)", "Length(BC,$2sqrt{3}$)" , | "Parallel(AB,DC,0)", "Parallel(AD,BC,0)", "Perpendicular(AH,BD,90)", |
| In triangle ABC, ∠ BCA=90 °, AC=3, ∠ ABC=30 °, point P is the moving point on the side of BC, connecting AP. | "Polygon(ABC)" | "PointAtLine(P,BC,0)" | "Length(AC,3)", "ConnectPoints(AP)" | "Perpendicular(CB,CA,90)", "Angle(ABC,30)" |
| AB is the diameter of ⊙ O, and the string CD ⊥ AB connects OC and BD at point E. ∠ AOC=110 °. | "Circle(O,r)", "StringOfCircle(O,AB)", "StringOfCircle(O,CD)" | "LineIntersect(AB,CD,E)" | "PointLineDistance(O,AB,0)", "ConnectPoints(OC)", "ConnectPoints(BD)" | "Perpendicular(AB,CD,90)", "Angle(AOC,110)" |
| In rhombus ABCD, the diagonals AC and BD intersect at point O, where OA=1 and OB=2. | "Rhombus(ABCD)" | "LineIntersect(AC,BD,O)" | "LengthRatio(AB,CD,Ratio(1,1))", "LengthRatio(BC,AD,Ratio(1,1))", "LengthRatio(AB,BC,Ratio(1,1))", "Length(OA,1)", "Length(OB,2)" | "Parallel(AB,DC,0)", "Parallel(AD,BC,0)" |
| In the rhombus ABCD, E, F, G, and H are the midpoints of AB, BC, CD, and DA, respectively. AB=6, ∠ ABC=60 °, connecting EF, FG, GH, and HE. | "Rhombus(ABCD)" | "PointAtLine(E,AB,2)", "PointAtLine(F,BC,2)", "PointAtLine(G,CD,2)", "PointAtLine(H,DA,2)" | "LengthRatio(AB,CD,Ratio(1,1))", "LengthRatio(BC,AD,Ratio(1,1))", "LengthRatio(AB,BC,Ratio(1,1))", "Length(AB,6)", "ConnectPoints(EF)","ConnectPoints(FG)","ConnectPoints(GH)", "ConnectPoints(HE)" | "Parallel(AB,DC,0)", "Parallel(AD,BC,0)", "Angle(ABC,60)" |
| In the rhombus ABCD, passing through point B is marked as BE ⊥ AD, BF ⊥ CD, with vertical feet at points E and F, connecting BD and extending to G, DG=BD, connecting EG and FG, AE=DE. | "Rhombus(ABCD)" | "DoPerpendicular(B,AD,E)", "DoPerpendicular(B,CD,F)", "ExtensionLine(BD,G)" | "LengthRatio(AB,CD,Ratio(1,1))", "LengthRatio(BC,AD,Ratio(1,1))", "LengthRatio(AB,BC,Ratio(1,1))", "LengthRatio(DG,BD,Ratio(1,1)),", "LengthRatio(AE,DE,Ratio(1,1))", "ConnectPoints(EG)", "ConnectPoints(FG)" | "Parallel(AB,DC,0)", "Parallel(AD,BC,0)" |

Figure 16: The formal language corresponding to natural language in Figure17.

| Natural Language | Initial state | Optimization process | | | | Final state |
|---|---|---|---|---|---|---|
| In △ ABC, ∠ ACB=90 °, ∠ ABC=30 °, the angle bisector AD of ∠ BAC passing through point A intersects BC at point D, and the perpendicular foot of DE ⊥ AB passing through point D is E, DE=1. | | | | | | |
| In △ ABC, ∠ ABC=90 °, AB=8, BC=12, D is a moving point on the AC side, connecting BD. E is a moving point on BD, connecting AE and CE, ∠ ABD=∠ BCE. | | | | | | |
| In △ ABC, ∠ ACB=90 °, AD divides ∠ BAC equally and intersects BC with D, DE is perpendicular to AB with E. DE=1.5, BD=3 | | | | | | |
| In the parallelogram ABCD, BD is the diagonal, BD=CD, ∠ BCD=70 °, AE is perpendicular to BD at point E. | | | | | | |
| In parallelogram ABCD, the diagonal lines AC and BD intersect at point O, AB ⊥ AC. Point H is on the BD, AH⊥BD , AB=2, BC=$2sqrt\{3\}$. | | | | | | |
| In triangle ABC, ∠ BCA=90 °, AC=3, ∠ ABC=30 °, point P is the moving point on the side of BC, connecting AP. | | | | | | |
| AB is the diameter of ⊙ O, and the string CD ⊥ AB connects OC and BD at point E. ∠ AOC=110 °. | | | | | | |
| In rhombus ABCD, the diagonals AC and BD intersect at point O, where OA=1 and OB=2. | | | | | | |
| In the rhombus ABCD, E, F, G, and H are the midpoints of AB, BC, CD, and DA, respectively. AB=6, ∠ ABC=60 °, connecting EF, FG, GH, and HE. | | | | | | |
| In the rhombus ABCD, passing through point B is marked as BE ⊥ AD, BF ⊥ CD, with vertical feet at points E and F, connecting BD and extending to G, DG=BD, connecting EG and FG, AE=DE. | | | | | | |

Figure 17: Visualization of optimization process of geometric diagrams.

# F USE OF LARGE LANGUAGE MODELS

In this work, large language models (LLMs) were utilized as auxiliary tools to support the rewriting and polishing of certain passages during manuscript preparation. All model-generated outputs were thoroughly examined and refined by the authors to ensure precision and compliance with academic standards. The overall research design, analysis, and conclusions were conceived and conducted independently by the authors. The authors bear full responsibility for the accuracy, originality, and integrity of the content presented in this paper.

