# OpenReview forum: "GeoLoom: High-quality Geometric Diagram Generation from Textual Input"
_ICLR.cc/2026/Conference — Submitted to ICLR 2026_

### Official Review · Reviewer_PrUm · 2025-10-26

**Soundness:** 4
**Presentation:** 4
**Contribution:** 4
**Rating:** 6
**Confidence:** 4

**Summary:**

This paper presents GeoLoom, a novel two-stage framework for generating high-quality, geometrically accurate diagrams from natural language descriptions. GeoLoom's approach is to first translate the unstructured natural language into a structured, formal representation, and then use a specialized solver to render the diagram from that representation.

**Strengths:**

1. Instead of trying to force a single, end-to-end model to understand both language and geometry, the authors intelligently divide the problem. They use LLMs for what they excel at (structured language translation) and a dedicated optimization algorithm for what it excels at (constraint satisfaction). This is a much more robust and interpretable design than a black-box diffusion model.

2. The design of a "generation-oriented" formal language is a key insight. Correctly identifying that languages for solving problems are not sufficient for generating them and encoding "constructive dependencies" (free vs. dependent points ) is a strong conceptual contribution.

3. Creating a new, manually verified dataset of 4,730 paired examples is a substantial effort. This dataset will be essential for benchmarking future work in this domain.

4. The visual comparisons in Figure 4 are definitive. GeoLoom produces clean, mathematically correct diagrams, while the state-of-the-art baselines produce unusable, garbled messes. This is strongly supported by the user study, where GeoLoom (especially the fine-tuned version) was preferred by a massive margin over AutomaTikZ and Seedream.

**Weaknesses:**

1. The entire pipeline's success hinges on the first step: correct autoformalization. However, the quantitative results in Table 1 show that the "True" accuracy (verified by manual examination) for the best fine-tuned model (Qwen2.5-7b) is only 85.34%. This implies that for ~15% of inputs, the formal language is incorrect from the start. If the GeoLingua representation is wrong, the coordinate solver will perfectly generate the wrong diagram.

2. The paper repeatedly uses the term "coordinate solver". However, the method described is a stochastic optimization algorithm (Monte Carlo), not a deterministic solver (like a symbolic or constraint-based solver). This approximate method is sensitive to hyperparameters (T, Q)  and may not be guaranteed to find the true global optimum, especially for highly complex diagrams. It could get stuck in a local minimum where constraints are still violated.

3. The current framework and solver are designed exclusively for 2D Euclidean geometry. The authors acknowledge this limitation in the conclusion. This means the system cannot handle 3D geometry or non-Euclidean spaces, which limits its application.

**Questions:**

1. The ~15% failure rate of the autoformalization step is the system's main bottleneck. Could the authors provide a more detailed analysis of these failures? What are the most common types of errors? Are they simple parsing mistakes (e.g., wrong length value), or deeper semantic misunderstandings (e.g., misinterpreting a "dependence" or "perpendicular" relationship)?

2. Why was a stochastic Monte Carlo optimizer chosen over a deterministic geometric constraint solver? While the MC method is fast, a deterministic solver could potentially find an exact solution without needing to tune iteration hyperparameters (T and Q)  or worrying about local minima. What are the advantages of the optimization approach that outweigh this?

---

> ### Author Response · Authors · 2025-11-27
> **Reply to Reviewer PrUm's comments**
>
> Thank you for your thoughtful comments and suggestions.
>
> Here we address the points you mentioned in the Weaknesses part.
>
> ****
> **Weakness1:** If the GeoLingua representation is wrong, the coordinate solver will perfectly generate the wrong diagram.
>
> **Response**: We acknowledge that errors in the autoformalization step can propagate to the generated diagrams. However, a detailed **analysis of failure cases(lines 413-419 of the revised manuscript)** indicates that most formal language errors arise from topological ambiguities in non-standard natural language descriptions (e.g., AB//CD being written as AB//DC). In such cases, the formal representation may misinterpret the intended orientation—for instance, treating AB and DC as a 0° angle, whereas the intended AB and CD are at 0°, and AB and DC are at 180°. Importantly, statistical analysis shows that only ~3% of generated diagrams exhibit structural errors directly attributable to formal language mistakes. The primary source of failure remains the local minimum problem of the coordinate solver. Consequently, the effective accuracy of formal representations for practical diagram generation is substantially higher than the manually verified "True" accuracy, mitigating the concern that initial autoformalization errors dominate the pipeline’s performance.
>
> ****
>
> **Weakness2:** "coordinate solver" could get stuck in a local minimum where constraints are still violated.
>
> **Response**: We appreciate the reviewer’s observation regarding the terminology. While our “coordinate solver” employs a Monte Carlo–based stochastic optimization, it is designed to efficiently satisfy geometric constraints and produce coordinates with high structural fidelity. To contextualize its effectiveness, **in Section 4.1(lines 314-318) and Appendix D (905-917) of the revised manuscript,** we compare our method against Penrose (a manually coded symbolic approach, representing a deterministic baseline) and GeoGebra (manual diagram drawing) on complex IMO-level problems. Although Penrose provides a fully deterministic solution, it requires extensive prior knowledge of symbolic languages, whereas our approach achieves comparable accuracy within practical thresholds with substantially less manual effort. Furthermore, **in Section 4.2(lines 413-419) and Appendix E (lines 967-1019) of the revised manuscript**, we include a detailed failure-case analysis highlighting instances where the solver may converge to local minima.
> ****
>
> **Weakness3**: The current framework and solver are designed exclusively for 2D Euclidean geometry. The authors acknowledge this limitation in the conclusion. This means the system cannot handle 3D geometry or non-Euclidean spaces, which limits its application.
>
> **Response**: We appreciate the reviewer’s observation. While GeoLoom is currently demonstrated on 2D Euclidean geometry, the underlying framework—an end-to-end text-to-diagram generation via formalization and constraint-based solving—is general and extensible. Non-Euclidean, 3D, and dynamic geometries would require adapting the formal language and solver, but the core methodology remains applicable. We highlighted these extensions as directions for future work.
>
> ****
>
> Then here are the answers to the Questions part.
>
> **Question1**: The ~15% failure rate of the autoformalization step is the system's main bottleneck. Could the authors provide a more detailed analysis of these failures?
>
> **Response**: Please refer to our response to Weakness 1.
>
> ****
>
> **Question2**: Why was a stochastic Monte Carlo optimizer chosen over a deterministic geometric constraint solver? While the MC method is fast, a deterministic solver could potentially find an exact solution without needing to tune iteration hyperparameters (T and Q) or worrying about local minima. What are the advantages of the optimization approach that outweigh this?
>
> **Response**: We thank the reviewer for the insightful question. Deterministic geometric constraint solvers, while theoretically capable of finding exact solutions, typically need to exhaustively traverse all possible coordinate combinations. This results in a large search space and high computational cost, especially for diagrams with many interdependent constraints or partially specified elements.
>
> In contrast, our GeoLoom framework employs a Monte Carlo (MC) stochastic optimizer, which offers several practical advantages. First, it is fast and efficient, allowing exploration of the solution space without exhaustive enumeration. Second, the stochastic approach naturally handles ambiguous scenarios by iteratively refining coordinates while balancing constraint satisfaction. Third, it scales well to complex diagrams, where deterministic solvers may become computationally infeasible or fail to converge.
>
> In practice, we find that the MC optimizer, with moderate iteration parameters, reliably produces geometrically accurate diagrams while maintaining efficiency and robustness.

---

### Official Review · Reviewer_yC6F · 2025-10-27

**Soundness:** 3
**Presentation:** 3
**Contribution:** 3
**Rating:** 6
**Confidence:** 3

**Summary:**

The authors introduce GeoLoom, a novel framework for generating high-precision geometric diagrams from natural language descriptions. The method features a two-stage architecture: (1) an autoformalization module that converts natural language into a new formal language called GeoLingua, and (2) a Monte Carlo–based coordinate solver that generates spatially accurate diagrams by minimizing constraint violations. To support this framework, the authors construct GeoNF, a benchmark dataset of 4,730 aligned natural language and formal specification pairs. Experiments use both training-free (e.g., DeepSeek-v3) and fine-tuned LLMs (LLaMA, Qwen) for the autoformalization stage. Evaluation includes geometric constraint metrics (LCI, ADI), human judgment, user studies, and generation speed. GeoLoom consistently outperforms strong baselines like AutomaTikZ and Seedream 3.0 in structural fidelity, efficiency, and user preference.

**Strengths:**

1. **Strong and novel method.** GeoLingua presents a principled framework for geometric diagram generation. This suggested architectural design is well-supported by strong empirical results: GeoLoom achieves top performance in constraint satisfaction metrics (LCI, ADI), manual accuracy, and user preference in both diagram quality and alignment.
2. **Clear writing and presentation.** The paper is clearly written, with informative diagrams, making the method easy to follow.

**Weaknesses:**

1. **Some geometric constraint are hard to understand.** The paper introduces several evaluation equations (e.g., Eq. 2 and Eq. 5 for length/angle relations) using Iverson bracket notation and conditional terms, but does not provide intuition or derivations. It’s unclear how these metrics correspond to geometric correctness or what theoretical justification supports their formulation.
2. **No ablation study on GeoLingua components or constraint types.** The impact of individual constraint types (e.g., length ratio vs angle value) on final diagram quality is not explored, missing an opportunity to better understand what aspects contribute most to structural fidelity.
3. **No extrinsic evaluation.** Beyond constraint satisfaction and human preference, the authors could evaluate their system’s practical utility in downstream tasks. For instance, one could give the generated diagrams as auxiliary inputs to VLMs solving geometry problems (e.g., GPT, Gemini) and measure whether models solve more problems or do so more accurately depending on the diagram source. This would provide a task-grounded measure of diagram quality and educational value.

**Questions:**

1. It may improve clarity to use consistent terminology between Section 2 (L140–141: “basic geometric primitives”, “point dependencies”, etc.) and the formal language components listed later (L146–156: “shapes”, “dependence”, etc.).
2. In Algorithm 2 line 17, what does the constant 50 mean used for canvas sizing?

---

> ### Author Response · Authors · 2025-11-27
> **Reply to Reviewer yC6F's comments**
>
> Thank you for your valuable feedback.
>
> Here we address the points you mentioned in the Weaknesses part.
> ****
> **Weakness1: Some geometric constraint are hard to understand.** The paper introduces several evaluation equations (e.g., Eq. 2 and Eq. 5 for length/angle relations) using Iverson bracket notation and conditional terms, but does not provide intuition or derivations. It’s unclear how these metrics correspond to geometric correctness or what theoretical justification supports their formulation.
>
> **Response**: We thank the reviewer for pointing out the need for clearer intuition behind the constraint formulations. We have clarified the definition of the loss function and the five geometric constraints, and the clarifications are now included in **Section 3.2 (lines 258-260, 270-283).**
>
> ****
>
> **Weakness2: No ablation study on GeoLingua components or constraint types.**
>
> **Response**: We thank the reviewer for the insightful suggestion regarding ablation studies on GeoLingua components. Following this advice, we conducted a detailed constraint-family ablation study to isolate the impact of individual constraint types on diagram quality. **The experimental settings and results are reported in Section 4.2(lines 429-450) and Table 4 of the revised manuscript.**
>
> Our findings highlight the distinct contributions of length and angle constraints:
>
> + **Length constraints only** achieve the lowest LCI (0.651) and the fastest convergence (2.844 s). By removing angular constraints, the optimization landscape becomes smoother and less restrictive, enabling the solver to enforce metric structure more efficiently and avoid local minima. This indicates that length constraints are highly stable, informative, and straightforward to optimize.
> + **Angle constraints only** maintain reasonable angular accuracy (ADI = 1.577), but incur substantially higher computational cost (7.477 s, nearly three times slower than length-only). Angle constraints introduce sharper non-linearities, making the solver more prone to local minima and sensitive to perturbations. This confirms that angle constraints, while crucial for geometric correctness, are inherently more difficult to satisfy during optimization.
>
> Overall, these results validate the reviewer’s observation: different constraint types contribute differently to structural fidelity, with length constraints providing stable guidance and angle constraints imposing more restrictive geometric conditions.
> ****
> **Weakness3: No extrinsic evaluation.** Beyond constraint satisfaction and human preference, the authors could evaluate their system’s practical utility in downstream tasks.
>
> **Response**: We sincerely thank the reviewer for the insightful suggestion. We note, however, that using downstream geometry problem-solving as a proxy for diagram quality has a fundamental limitation: many geometry problems can be solved purely symbolically from the text description without requiring comprehension of the diagram.
>
> To illustrate this, we tested three complex problems from the IMO Shortlist 2014, APMO 2017, and EGMO/AoPS, each involving multiple auxiliary points, angle. Remarkably, GPT-4o solved all of these problems purely from textual reasoning, without any diagram. This demonstrates that problem-solving success does not reliably reflect the correctness or utility of the associated geometric diagram.
>
> Consequently, we argue that diagram correctness should be evaluated structurally, as in our proposed metrics (LCI, ADI, angle relations, length ratios), which directly quantify geometric fidelity. **For transparency, we include the three illustrative examples in the supplementary materials to clarify the limitations of problem-solving-based validation.**
>
> ****
> Then here are the answers to the Questions part.
>
> **Question1:** It may improve clarity to use consistent terminology between Section 2 (L140–141: “basic geometric primitives”, “point dependencies”, etc.) and the formal language components listed later (L146–156: “shapes”, “dependence”, etc.).
>
> **Response**: We thank the reviewer for this helpful suggestion. To improve clarity and consistency, we have standardized the terminology between Section 2 and the description of formal language components **(Line 140, 146-156). The revised terms are now aligned throughout the manuscript.**
> ****
> **Question2:** In Algorithm 2 line 17, what does the constant 50 mean used for canvas sizing?
>
> **Response**: The constant 50 in Algorithm 2 (line 17) is used to ensure that the canvas size is always an integer multiple of 50. This design choice provides consistent scaling across generated diagrams, simplifying coordinate placement and avoiding fractional pixel issues. **For clearer expression, we have provided a detailed explanation in line 18 of Algorithm 2 in the revised version.**

---

### Official Review · Reviewer_FpEe · 2025-10-31

**Soundness:** 3
**Presentation:** 2
**Contribution:** 3
**Rating:** 4
**Confidence:** 3

**Summary:**

The paper proposes GeoLoom, which converts natural language geometric descriptions into constrained coordinates through a newly designed formal language called GeoLingua. Then, Monte Carlo optimization is used to generate high-precision two-dimensional geometric diagrams, and a paired dataset called GeoNF and a structural deviation index are constructed. Experimental results show that its structural fidelity is superior to baselines such as Seedream3.0 and AutomaTikZ, and most samples are completed within 10 seconds.

**Strengths:**

1.For the first time, the "formal language + symbolic solving" paradigm of geometric problem-solving has been transferred to image generation, presenting a novel approach.

2.The self-developed GeoLingua explicitly encodes the construction sequence and constraints, facilitating subsequent coordinate calculations.

3.Quantifiable indicators such as LCI/ADI have been proposed to provide an objective evaluation benchmark for the community.

4.The two-stage process supports two modes: training-free prompts and fine-tuning, which is flexible and easy to use.

**Weaknesses:**

1.This is only applicable to 2D Euclidean geometry. Non-Euclidean, 3D or dynamic geometry need to be rewritten for constraints and solvers.

2.Monte Carlo random sampling relies on a large number of iterations, often taking over 50 seconds for complex graphs, and the convergence is slow.

3.The grammar of formal languages is fixed. If the description contains advanced concepts such as "similar" or "trajectory", it cannot be expressed.

4.The dataset size is still small, and it only comes from middle school questions. It is insufficient in covering university and competition-level curves and solid geometry.

5.There is no comparison with interactive drawing software (such as GeoGebra), and only compared with the generated model, the persuasiveness is limited.

6.The robustness under noisy text (oral, multilingual) has not been evaluated. Actual classroom input often contains typos.

7.The indicators LCI/ADI only measure relative deviation and are insensitive to topological errors (such as reversed point order), which may be overestimated.

8.No failure sample analysis is provided. Readers cannot know the system boundary and failure modes.

**Questions:**

see weakness

---

> ### Author Response · Authors · 2025-11-27
> **Reply to Reviewer FpEe's comments about Weakness1-Weakness4**
>
> Thank you for your constructive comments.
>
> Here we address the points you mentioned in the Weaknesses part.
> ****
> **Weakness1:** This is only applicable to 2D Euclidean geometry. Non-Euclidean, 3D or dynamic geometry need to be rewritten for constraints and solvers.
>
> **Response**: We appreciate the reviewer’s observation. While GeoLoom is currently demonstrated on 2D Euclidean geometry, the underlying framework—an end-to-end text-to-diagram generation via formalization and constraint-based solving—is general and extensible. Non-Euclidean, 3D, and dynamic geometries would require adapting the formal language and solver, but the core methodology remains applicable. We highlighted these extensions as directions for future work.
>
> ****
>
> **Weakness2:** Monte Carlo random sampling relies on a large number of iterations, often taking over 50 seconds for complex graphs, and the convergence is slow.
>
> **Response:** We appreciate the reviewer’s concern regarding the runtime of Monte Carlo optimization. **We evaluated GeoLoom against two baselines on IMO-level geometric problems in the Section 4.1 (lines 313-318) and Appendix D (lines 905-917) in the revised manuscript**: the Penrose method, which requires manually authoring symbolic representations, and the Geogebra approach, which relies on fully manual diagram construction. While it is true that generating highly complex diagrams may take over 50 seconds, GeoLoom consistently achieves substantially lower time costs compared to these baselines. Moreover, GeoLoom’s runtime remains practical for most geometric problem instances, and its efficiency can be further improved with parallelization or adaptive sampling strategies.
>
> ****
>
> **Weakness3:** The grammar of formal languages is fixed. If the description contains advanced concepts such as "similar" or "trajectory", it cannot be expressed.
>
> **Response**: While it is true that the grammar of a formal language is fixed at any given time, fixed grammars are a standard and necessary design choice in geometric reasoning systems. Prior work, including Google’s AlphaGeometry and its successor AlphaGeometry2, likewise employs a predefined formal language and incrementally extends its expressivity as the task scope grows. Our GeoLingua design follows the same principled paradigm: it provides a stable, well-defined grammar tailored to the current task of geometric diagram generation, enabling precise parsing, deterministic coordinate solving, and interpretable constraint construction. The reviewer’s concern regarding advanced geometric notions such as _similarity_ or _trajectory_ is well taken. These higher-level constructs are indeed beyond the scope of the current version of GeoLingua, but they are not fundamentally incompatible with our framework. On the contrary, the modularity of GeoLingua and our coordinate-solver pipeline is intentionally designed to allow progressive expansion.
>
> ****
>
> **Weakness4:** The dataset size is still small, and it only comes from middle school questions. It is insufficient in covering university and competition-level curves and solid geometry.
>
> **Response:** We appreciate the reviewer’s concern regarding dataset breadth. To address this, we have expanded **Section 4.1（lines 313-318） in the revised manuscript** to include additional qualitative experiments on IMO-level problems featuring substantially more intricate geometric configurations, including higher-order curves and multi-step constructions. These results show that GeoLoom, despite being trained on a modest middle-school–level dataset, generalizes effectively to significantly more complex university and competition-level scenarios, producing diagrams that preserve both structural constraints and geometric fidelity.

---

> ### Author Response · Authors · 2025-11-27
> **Reply to Reviewer FpEe's comments about Weakness5-Weakness8**
>
> **Weakness5:** There is no comparison with interactive drawing software (such as GeoGebra), and only compared with the generated model, the persuasiveness is limited.
>
> **Response:** We thank the reviewer for the suggestion. Interactive systems such as GeoGebra and declarative frameworks like Penrose represent a different problem setting: both require substantial human input—either step-wise construction (GeoGebra) or manual authoring of a full symbolic specification (Penrose). They do not parse natural language or autonomously infer geometric relations, and therefore cannot be evaluated on our end-to-end text-to-diagram benchmark. A quantitative comparison is thus not applicable.
>
> **To clarify this distinction, we add a brief qualitative comparison with GeoGebra and Penrose, in Section 4.1 (lines 313-318) and Appendix D (lines 905-917) in the revised manuscript.** As shown in **Figure5**, while these systems can produce correct diagrams with human effort, GeoLoom generates comparable diagrams fully automatically and within ~60 seconds. This highlights why interactive tools, though useful, are not direct baselines for the automated generation task we address.
>
> ****
>
> **Weakness6:** The robustness under noisy text (oral, multilingual) has not been evaluated. Actual classroom input often contains typos.
>
> **Response:** We appreciate the reviewer’s suggestion to examine robustness under noisy or imperfect text. Our dataset is sourced from real mathematical problems, which naturally contain several forms of linguistic variability—including colloquial phrasing (e.g., “AB and CD being the same length”), notation diversity (e.g., “triangle ABC”，“△ABC”), and even topologically inconsistent expressions (e.g., describing segment _AB_ as _BA_). These variations introduce noise that closely reflects authentic classroom input.
>
> Our failure-case analysis indicates that the autoformalization module is largely robust to colloquial or heterogeneous descriptions: such noise does not affect the correctness of the generated GeoLingua programs. The only cases that lead to systematic errors are topological-order inconsistencies, which cause incorrect constraint mapping. Following the reviewer’s suggestion, **we have expanded the discussion in Section 4.2 and highlighted these failure patterns**, clarifying the types of natural-language noise that GeoLoom already handles effectively and those that remain challenging.
>
> ****
>
> **Weakness7:** The indicators LCI/ADI only measure relative deviation and are insensitive to topological errors (such as reversed point order), which may be overestimated.
>
> **Response:** We appreciate the reviewer’s concern regarding the potential insensitivity of LCI/ADI to topological errors. In our setting, however, such topological ambiguities are eliminated _prior_ to metric computation through the validation filtering. GeoLingua enforces an explicitly ordered and semantically grounded representation of geometric relations. For instance, an expression such as `Angle(ABC, 60)` is parsed into a well-defined mathematical entity: the angle formed by vectors BA and BC. The parser guarantees the correct orientation and reference of each constituent segment. Consequently, the coordinate solver operates on a topology-consistent constraint graph, and LCI/ADI reflects only geometric deviation rather than topological noise.
>
> ****
>
> **Weakness8:** No failure sample analysis is provided. Readers cannot know the system boundary and failure modes.
>
> **Response:** In the **revised manuscript, we have added a systematic analysis of representative failure modes in Section 4.2 (lines 413-419) and Appendix E (967-1018).** The analysis delineates the system’s boundaries and characterizes its primary sources of error. Specifically, we identify three dominant categories of failure: (1) topological inconsistencies caused by incorrect or incomplete GeoLingua autoformalization, (2) suboptimal convergence of the Monte Carlo coordinate solver due to local minima, and (3) rendering artifacts such as unintended occlusion or overlap that impair visual fidelity.

---

### Official Review · Reviewer_Wd9S · 2025-11-01

**Soundness:** 2
**Presentation:** 3
**Contribution:** 2
**Rating:** 2
**Confidence:** 4

**Summary:**

This paper introduces **GeoLoom**, a novel two-stage framework for generating high-quality, geometrically accurate diagrams from natural language descriptions. The authors identify that existing text-to-image models fail at this task due to a lack of spatial precision.

The GeoLoom framework consists of:

1.  An **autoformalization module** that translates the natural language input into a newly proposed, generation-oriented formal language called **GeoLingua**.
2.  A **coordinate solver** that takes the formal constraints from GeoLingua and uses **Monte Carlo optimization** to find a set of precise coordinates that satisfy these constraints.

To support this framework, the authors also introduce the **GeoNF dataset**, which aligns natural language descriptions with their corresponding GeoLingua representations. The paper demonstrates that GeoLoom significantly outperforms baselines like AutomaTikZ and Seedream in structural fidelity, as measured by both automated metrics and human evaluation.

**Strengths:**

- **Significance & Motivation:** The paper tackles a well-defined and significant problem. The authors correctly observe that general-purpose text-to-image models are unsuitable for domains requiring high structural and spatial accuracy, such as mathematical diagrams. An automated tool for this task would have clear applications in education, research, and engineering.
- **Methodological Clarity:** The proposed two-stage, "parse-then-solve" pipeline is logical, interpretable, and well-structured. Separating the natural language understanding (autoformalization) from the geometric constraint satisfaction (coordinate solver) is a strong and principled design choice, drawing inspiration from advances in formal geometry problem-solving. The overall framework is clearly illustrated in Figure 2.
- **Novel Contributions:** The paper provides two valuable resources to the community:
  1.  **GeoLingua:** A formal language designed specifically for _generative_ tasks, which notably encodes constructive dependencies, not just static constraints.
  2.  **GeoNF Dataset:** A new dataset of 4,730 paired natural language and formal language descriptions , which addresses a key bottleneck of high-quality data for this task.
- **Strong Evaluation:** The authors perform a comprehensive evaluation, including:
  - Quantitative metrics (LCI, ADI) derived directly from constraint violations .
  - Manual verification of accuracy ("True" accuracy in Table 1).
  - A qualitative comparison (Figure 4) that clearly shows the limitations of baselines.
  - A human user study that measures both image quality and textual alignment.

**Weaknesses:**

Despite its strengths, the paper suffers from several weaknesses, primarily concerning the novelty of the paradigm, the scalability of the solver, and a lack of depth in key areas.

- **Missing Comparison to Key Baselines:** The core idea of a text-to-diagram pipeline based on formal language and constraint-based optimization is not new. The related work discusses GANs/diffusion and vector graphics generation, but omits a critical category: constraint-based diagram generation systems. A well-known example is **Penrose** (Penrose: From Mathematical Notation to Beautiful Diagrams, SIGGRAPH 2020), which also uses a formal language and stochastic optimization to generate diagrams. A comparison to this and similar systems is essential to properly contextualize GeoLoom's novelty and performance.
- **Scalability of the Monte Carlo Solver:** The choice of a Monte Carlo (MC) based solver raises significant concerns about scalability.
  - **Efficiency:** MC methods are stochastic and can be very inefficient in high-dimensional or highly-constrained search spaces. The paper claims "superior computational efficiency" and shows that _most_ examples are solved in <10 seconds (Table 3), but this does not constitute a proper scalability analysis. The efficiency will likely degrade exponentially as the number of points and constraints increases.
  - **Robustness:** The solver relies on probabilistically sampling to find a global optimum and avoid "suboptimal configurations". However, the paper provides no discussion of failure cases or an analysis of how often the solver gets stuck in a poor local minimum.
- **Vagueness in Loss Function Details:** The paper defines five types of geometric constraints and a final objective function. It is not clear how these five distinct metrics, which have different forms (ratios, Iverson brackets), are normalized and combined into the single set used in the final loss. For instance, "C_{lin\\_rel}" does not appear to be normalized around 1. This lack of clarity hinders reproducibility.
- **Oversimplified Evaluation Cases:** The examples shown in the qualitative evaluations (Fig. 4, 5, 9) are relatively simple geometric figures (triangles, quadrilaterals). The paper does not demonstrate that GeoLoom can handle highly complex problems with many interacting constraints, such as those found in geometry olympiads (e.g., IMO problems). This reinforces the concerns about the solver's scalability and robustness.
- **Lack of Ablation on Autoformalization:** The system's quality is highly dependent on the correctness of the autoformalization step. The paper mentions a "validation-based filter" to ensure syntactic correctness, but it does not discuss how semantic errors (e.g., misinterpreting a complex relationship) would propagate and affect the final diagram.

**Questions:**

1.  **Comparison to Penrose:** Could the authors elaborate on the novelty of GeoLoom compared to existing constraint-based diagram generation systems like Penrose, which also use a formal language and optimization-based solver?
2.  **Solver Scalability:** How does the Monte Carlo solver's performance (both in time and accuracy) scale with an increasing number of points and constraints (e.g., 5, 10, 20, 50 constraints)? The examples shown are simple; have the authors tested GeoLoom on more complex, "olympiad-level" geometry problems?
3.  **Solver Robustness (Local Minima):** Could the authors provide examples of failure cases where the MC solver gets stuck in a poor local minimum and fails to satisfy all constraints? How frequently does this occur, and are there any mechanisms besides random restarts to handle it?
4.  **Loss Function Clarification:** Could the authors please provide a more precise formulation of the objective function $\mathcal{L}(S)\_{max}$? Specifically, how are the five different constraint deviation metrics (e.g., "C_{lin\\_rat}" and "C_{lin\\_rel}" ) normalized and combined into the set $\mathcal{C}$?
5.  **Handling of Fully-Constrained Systems:** The solver is described as "iteratively perturbs the coordinates of free points ($P_f$)". What happens in the case of a diagram that is fully constrained or over-constrained and has no "free points" to perturb? How is the optimization initialized in such a case?
6.  **Error Propagation:** What is the impact of semantic errors from the autoformalization module? If the LLM generates a syntactically valid but _semantically incorrect_ formal description (e.g., `Perpendicular` instead of `Parallel`), the solver would presumably generate a "correct" diagram for the _wrong_ problem. How is this type of error handled or measured?

---

> ### Author Response · Authors · 2025-11-27
> **Reply to Reviewer Wd9S's comments  about Weakness1-Weakness3**
>
> Thank you for taking the time to review our paper.
>
> Here we address the points you mentioned in the Weaknesses part.
>
> **Weakness1：Missing Comparison to Key Baselines (Penrose).**
>
> **Response:** We appreciate the reviewer’s suggestion to discuss constraint-based systems such as _**Penrose**_. While Penrose is an important declarative diagramming framework, it requires users to manually author full symbolic specifications in its domain-specific language; it does not parse natural language or infer geometric relations. GeoLoom, in contrast, performs _end-to-end_ natural-language → formal specification → coordinate solving without human intervention. Because Penrose cannot operate on natural-language inputs, it cannot be evaluated on our benchmarks, making direct comparison infeasible.
>
> Nevertheless, we acknowledge the value of a qualitative comparison to clarify the distinction. **In the revised manuscript，Section 4.1(lines 313-318 ) and Appendix D(lines 905-917), we include a comparison with Penrose** (Ye et al., 2020) and GeoGebra (Hohenwarter, 2007), both of which require human intervention. All systems are able to generate correct diagrams **on IMO-style items (Fig. 5)**, but GeoLoom achieves this fully automatically and within ~60 seconds per diagram. In contrast, Penrose demands substantial user effort to translate each problem into its symbolic language. **We have added the research about constraint based geometry generation systems in the Text to Image Generation section of our related work(lines 501-505), to better situate GeoLoom within this literature.**
>
> Finally, the novelty of GeoLoom does not lie in the use of optimization alone, but in its _end-to-end integration_ of (1) autoformalization into a generation-oriented formal language, (2) Monte Carlo–based deterministic coordinate solving tailored to geometric constraints, and (3) a paired natural-language–formal dataset with a structural-deviation metric that enables supervised and iterative refinement.
> ****
>
> **Weakness2: Scalability of the Monte Carlo Solver:** The choice of a Monte Carlo (MC) based solver raises significant concerns about scalability.
>
> **Response:** We appreciate the reviewer’s concerns regarding the scalability and robustness of the MC-based solver. To address this, **we have added a comprehensive failure-case analysis in Section 4.2（lines 413-419）and Appendix E (lines 967-1018).**
>
> Our analysis evaluates all test-set failures from both qualitative and quantitative perspectives (Figure 13). A sample is counted as a failure if either autoformalization or coordinate solving produces an incorrect result. We find that:
>
> + **Autoformalization errors (≈3%)** arise from irregular or ambiguous natural-language descriptions, which lead to incorrect topological ordering in GeoLingua and consequently make the diagram unsatisfiable.
> + **Local-minimum solutions** are the primary solver-related failures. Due to the inherently non-convex constraint landscape, MC optimization may converge to suboptimal configurations even when the formal specification is correct. These cases account for 8% of failures for Qwen2.5-7B and about 12% on the full benchmark.
> + **Overlapping issues (≈3.5%)** occur when two points become excessively close. Although metric scores remain acceptable, the resulting diagrams are visually unsatisfactory.
>
> Regarding the question of alternative mechanisms beyond random restarts: at present, random restarts offer the most straightforward means of escaping poor local minima, though at the cost of additional computation. We agree that more principled approaches could further enhance robustness. One promising direction is initialization via retrieval, where a retrieval-augmented mechanism supplies coordinates from structurally similar diagrams as informed initialization rather than relying solely on random seeds. We view this as a valuable extension and will explore it in future work.
>
> ****
>
> **Weakness3: Vagueness in Loss Function Details:** "C_{lin\_rel}" does not appear to be normalized around 1. This lack of clarity hinders reproducibility.
>
> **Response:** We thank the reviewer for highlighting this important point. To address the concern, we now detail the normalization procedures applied to each constraint—ensuring comparable scales despite differing forms such as ratios and Iverson brackets—and the precise manner in which they are combined. **These revisions are provided in Section 3.2 (lines 258-260 and 270–282).**
> ****

---

> ### Author Response · Authors · 2025-11-27
> **Reply to Reviewer Wd9S's comments  about Weakness4-Weakness5**
>
> **Weakness4: Oversimplified Evaluation Cases**
>
> **Response:** We appreciate the reviewer’s concern regarding problem complexity. To address this, we have conducted experiments on IMO-level geometric problems. **Detailed settings and results are reported in Section 4.1(lines 313-318) and Figure 5 in the revised manuscript.** These results demonstrate that GeoLoom effectively handles complex diagrams with multiple interacting constraints, validating both the scalability and robustness of our coordinate solver beyond simple geometric figures.
> ****
>
> **Weakness5: Lack of Ablation on Autoformalization.**
>
> **Response:** Thank you for raising this important point.
>  We sincerely thank the reviewer for pointing out the need to more clearly justify the role of **validation filters**. **In the revised manuscript, we have added a dedicated ablation study (lines 420-426, Table 4) to directly address this concern**.
>
> These results demonstrate, adding validation filters to the training-free autoformalization substantially improves spatial accuracy: LCI decreases from **1.340 → 0.962** and ADI from **1.864 → 1.747**, leading to a notable improvement of the overall metric (1.602 → 1.337).
> More importantly, the filtered auto-formalization becomes **very close to manually annotated (“Labeled”) formal language**, whose full-constraint score is 1.203.
> This demonstrates that the proposed filters can effectively correct non-standard or noisy constraints generated during auto-formalization, ensuring that most structural errors are eliminated before coordinate optimization. This directly confirms their necessity and positive impact.

---

> ### Author Response · Authors · 2025-12-01
> **Reply to Reviewer Wd9S's comments about questions**
>
> Then here are the answers to the Questions part.
>
> **Question1: Comparison to Penrose.**
>
> **Response**: Please refer to our response for Weakness 1.
> ****
>
> **Question2: Solver Scalability:** How does the Monte Carlo solver's performance (both in time and accuracy) scale with an increasing number of points and constraints (e.g., 5, 10, 20, 50 constraints)? The examples shown are simple; have the authors tested GeoLoom on more complex, "olympiad-level" geometry problems?
>
> **Response:** Please refer to our response for Weakness 4.
> ****
>
>
> **Question3: Solver Robustness (Local Minima).**
>
> **Response:** Please refer to our response for Weakness 2.
>
> ****
>
> **Question4: Loss Function Clarification:** Could the authors please provide a more precise formulation of the objective function ? Specifically, how are the five different constraint deviation metrics (e.g., "C_{lin\_rat}" and "C_{lin\_rel}" ) normalized and combined into the set ?
>
> **Response:** Please refer to our response for Weakness 3.
> ****
>
>
> **Question5: Handling of Fully-Constrained Systems:** The solver is described as "iteratively perturbs the coordinates of free points ()". What happens in the case of a diagram that is fully constrained or over-constrained and has no "free points" to perturb? How is the optimization initialized in such a case?
>
> **Response:** In practice, a geometric diagram may contain dependent points, but it always includes free points that define its structure. For instance, a simple line segment AB has no dependent points; the endpoints A and B serve as free points. More generally, even in fully-constrained diagrams, the solver identifies an initial set of free points whose coordinates determine the configuration of all dependent points. Optimization is therefore always initialized on these free points, ensuring that the system can be solved iteratively.
>
> ****
>
> **Question6: Error Propagation:**  What is the impact of semantic errors from the autoformalization module? If the LLM generates a syntactically valid but semantically incorrect formal description (e.g., Perpendicular instead of Parallel), the solver would presumably generate a "correct" diagram for the wrong problem. How is this type of error handled or measured?
>
> **Response:** We thank the reviewer for raising this important point. In practice, semantic errors from the autoformalization module are rare. **In Section 4.2(lines 413-418) and Appendix E(lines 967-1018) in the revised manuscript,** we provide a detailed failure case analysis, identifying three primary sources of errors: (i) non-standard natural language descriptions leading to syntactic errors in formalization, (ii) Monte Carlo solver convergence to local minima, and (iii) visual overlap of points in the generated diagram. Importantly, the majority of errors in the formal language arise from syntactic issues rather than semantic misinterpretation (**as illustrated in Figure 13 of Appendix E**), which can cause extreme abnormalities in LCI and ADI indicators and thus be measured.

---

### Author Response · Authors · 2025-12-03
**Rebuttal Summary (Part One) to the Newly Assigned Area Chair**

To the newly assigned Area Chair,

Since reviewers can no longer join discussions or update scores, we sincerely thank them for their initial evaluations, which greatly helped improve our manuscript. We are especially grateful to the newly assigned Area Chair for taking on this responsibility. To support evaluation, we have included a “Rebuttal Summary” below, outlining the main revisions and responses addressing the reviewers’ concerns.

This paper introduce **GeoLoom**, a text-to-diagram framework that converts natural language into a formal geometric representation (**GeoLingua**) and generates precise diagrams using a **Monte-Carlo coordinate solver**. We also introduced the **GeoNF dataset** with alinged natural language and formal descriptions and for structural supervision. Experiments show that GeoLoom delivers **higher accuracy and better computational efficiency** than existing methods.

****

**Key strengths highlighted by reviewers include:**

+ Clear significance of the problem and motivation (Wd9S, FpEe)
+ Strong and novel method (yC6F, FpEe), intelligent and principled designed two-stage pipeline (Wd9S, PrUm)
+ Two valuable resources of Geolingua (formal language) and GeoNF dataset (paired natural language and formal language) to the community (Wd9S, FpEe, PrUm)
+ The value of quantifiable indicators (FpEe)
+ Strong Evaluations (Wd9S, PrUm)
+ Clear writing and presentation (yC6F)

****

**In response to reviewer concerns, we provided the following clarifications and additional results:**

+ **Comparison with constraint-based systems (Wd9S, FpEe, PrUm):** We thank the reviewer for the suggestion. Penrose is a powerful constraint-based diagramming system, but it requires users to manually author symbolic specifications and cannot parse natural language; GeoGebra is a completely handmade drawing system. GeoLoom instead performs fully automatic natural language → formal specification → coordinate solving. To clarify this distinction, we integrated related constraint-based generation research into the revised **related work section(lines 501-505),** to better situate GeoLoom within this literature. We  also added qualitative comparisons with **Penrose** and **GeoGebra** in **Section 4.1(lines 313-318 ) and Appendix D (lines 905-917).** The comparison **results are shown in Figure 5,** GeoLoom can handle the problems solved by Penrose and GeoGebra, and has a time advantage. Finally, GeoLoom’s novelty lies not in optimization alone but in its end-to-end pipeline: automatic formalization, constraint-guided Monte-Carlo solving, and a paired natural language–formal dataset enabling supervised refinement.

+ **Results on IMO-style diagram generation (Wd9S, FpEe):** We highly appreciate the reviewer's suggestion to add complex cases. GeoNF's geometry problems come from high school geometry problems. To verify the robustness of the GeoLoom method, we compared it on IMO style questions with the Penrose and GeoGebra methods that require manual participation. **As shown in Figure 5**, all systems can produce correct IMO-style diagrams, and GeoLoom does so without human intervention within ~60s.

---

> ### Author Response · Authors · 2025-12-03
> **Rebuttal Summary (Part Two) to the Newly Assigned Area Chair**
>
> + **Failure analysis (Regarding the scalability of the solver) (Wd9S, FpEe, PrUm):** In the **revised manuscript**, we have added a systematic analysis of representative failure modes **in Section 4.2 (lines 413-419) and Appendix E (967-1018)**. The analysis delineates the system’s boundaries and characterizes its primary sources of error. Specifically, we identify three dominant categories of failure:  **(1)Autoformalization errors (~3%)** stem from ambiguous or irregular natural-language descriptions, causing incorrect topological ordering in GeoLingua and unsatisfiable constraints. **(2)Local minima** are the main solver failures, where the non-convex landscape causes MC optimization to converge to suboptimal diagrams despite correct formal inputs. **(3)Point-overlap issues (~3.5%)** arise when two points become too close, yielding acceptable metric scores but visually poor diagrams. Regarding the local minima: at present, random restarts offer the most straightforward means of escaping poor local minima, though at the cost of additional computation. We agree that more principled approaches could further enhance robustness. One promising direction is initialization via retrieval, where a retrieval-augmented mechanism supplies coordinates from structurally similar diagrams as informed initialization rather than relying solely on random seeds. We view this as a valuable extension and will explore it in future work.
>
>
> + **Ablations on Autoformalization (Wd9S):** We thank the reviewer for highlighting this important issue. To address it, we added a dedicated ablation study **in Section 4.2(lines 420–426) and Table 4 of revised manuscript**. The results show that validation filters markedly improve auto-formalization quality, reducing **LCI from 1.340 → 0.962** and **ADI from 1.864 → 1.747**, and improving the overall metric from **1.602 → 1.337**. Notably, the filtered auto-formalization becomes highly comparable to the manually annotated (Labeled) formal language with a full-constraint score of 1.203. This confirms that validation filters effectively correct noisy or non-standard constraints and are essential for producing reliable formal inputs for the coordinate solver.
>
>
> + **Ablations on Geolingua components (yC6F):** We thank the reviewer for the helpful suggestion. In response, we performed a constraint-family ablation to isolate the roles of length and angle constraints **in Section 4.2 (lines 429-450) and Table4 of  revised manuscript**. The results reveal clear differences: **length  constraints  only** settings yield the lowest LCI (0.651) and fastest convergence (2.844 s), indicating that length constraints are stable, informative, and easy to optimize. In contrast, **angle  constraints  only**< settings preserve reasonable angular accuracy (ADI = 1.577) but incur much higher computational cost (7.477 s) due to stronger non-linearities and susceptibility to local minima. Overall, the ablation confirms that length constraints offer smoother, more reliable guidance, whereas angle constraints are more restrictive and harder to satisfy, aligning with the reviewer’s insight.
>
>
> + **Loss function Details (Wd9S, yC6F):** We thank the reviewer for highlighting this important point. To address the concern, we now detail the normalization procedures applied to each constraint ensuring comparable scales despite differing forms such as ratios and Iverson brackets, and the precise manner in which they are combined. **These revisions are provided in Section 3.2 (lines 258-260 and 270–282).**
>
>
> + **Extension beyond 2D Euclidean geometry (FpEe, PrUm):** We thank the reviewer for the comment. Although GeoLoom is demonstrated in 2D Euclidean geometry, the overall text-to-diagram pipeline is general. Extending to non-Euclidean, 3D, or dynamic geometries mainly requires adapting the formal language and solver, while the core methodology remains applicable. We note these as future work directions.
>
>
> The reviewers' insightful feedback mainly regarding experimental details, particularly robustness in complex cases, failure case analysis, and ablation studies. We have carefully responded with additional experiments and clarifications, further enhancing the rigor and completeness of our work.

---

### Meta-Review · Area_Chair_AnrM · 2025-12-26

**Summary:**

The paper tackles the problem of generating geometry diagrams from text.  To do so, it proposes a geometric formal language GeoLingua and dataset GeoNF of 4,730 geometric diagrams expressed using GeoLingua and paired witha natural language description.  The work also proposes GeoLoom, a two-stage framework for generating geometric diagrams that consists of 1) an autoformalization module that uses an LLM to convert natural language into structured specification in GeoLingua, and 2) a coordinate solver that optimizes position of diagram elements to satisfy specified constraints.

Reviewers appreciated the problem formulation, noting that the proposed two-stage pipeline is logical and interpretation, the contributions of both GeoLingua and GeoNF, as well as strong evaluation.

At the same time, reviewers also had a number of concerns (see "Reviewer Concerns" for details.). The AC finds that the key concerns are only partially addressed, and thus does not recommend the work for acceptance.

In particular, the necessity for the GeoLingua language is unclear to the AC, and the need for comparisons against prior work was not fully addressed.
- There should be quantitative comparison against important prior work such as Penrose.  While the authors argue that that Penrose "requires full symbolic specifications in its domain-specific language", a LLM prompting based approach to convert from natural language to the Penrose DSL would be natural.
- In addition, as a key component of GeoLoom is the introduction of a DSL of its own (GeoLingua), the need for a new geometric formal language should be established more.  For instance, it should be made clearer how does GeoLingua differ from the language used by Penrose.  What are the advantages of GeoLingua over existing languages (potentially a closer match to how concepts are expressed in natural language)?
- The AC also recommends the related work be pulled up to be after the introduction.

**Reviewer Concerns:**

Reviewers noted the following concerns.  For each, the AC note whether the concern was addressed or not.
1. Missing comparison to key baselines (Wd9S,FpEe)
   - Reviewer noted prior work Penrose (Wd9S) and GeoGebra (FpEe) were not compared against
   - *Partially addressed by author response*
     Qualitative comparisons added, but no quantitative results. The authors argue that Penrose requires the specification of symbolic specification vs natural language.  However, the AC believes it would be natural to have a training-free LLM  based approach to translate from natural language to the Penrose symbolic specification as a point of comparison.  The work can also benefit from more discussion about how it differs from MagicGeo and whether MagicGeo should also be compared against.
2. Missing ablations (Wd9S, yC6F)
   - Lack of ablation on autoformalization (Wd9S)
   - Lack of ablation on GeoLingua components (yC6F)
   - *These ablations were added to the manuscript*
3. Oversimplified evaluation cases (Wd9S)
   - Small dataset, limited to middle school questions (FpEe)
   - *Partially addressed by adding qualitative examples for diagrams from IMO dataset* Dataset is still limited.
4. Lack of failure analysis (FpEe)
   - *Addressed with failure analysis for Qwen2.5-7B experiments*
5. Limitations of GeoLingua (FpEe)
   - Limitations in coverage
   - Some geometric constraints are hard to understand
   - *Partially addressed by adding more details for the geometric constraints* Unclear if discussion of limitations of the coverage of GeoLingua was added to the manuscript.
6. The autoformalization module as a bottleneck / source of error progation (Wd9S, PrUm)
   - *Addressed with discussion of failure analysis*
7. Concerns regarding the Monte Carlo solver in the "coordinate solver" (Wd9S, FpEe)
   - Questions about the robustness and scalability of the solver (Wd9S, FpEe)
   - Whether the method can handle a fully-constrained system (Wd9S)
   - Questions about choice of MC solver over deterministic geometric constraint solver (PrUm)
   - *Mostly addressed with qualitative example on IMO data and discussion in failure analysis* Response was provided about the choice of MC solver over deterministic geometric constraint solver, but it is unclear if the manuscript was updated to describe the rationale for the design choice.
8. Robustness to noisy text (FpEe)
   - *Addressed with discussion in Section 4.2 (failure analysis)*
9. Concerns about metrics being insensitive to topological error (FpEe)
   - *Partially addressed* Authors provided response that in their setting, their method will filter out these topological ambiguities.  It is not clear whether the metric would be robust to a different approach where such ambiguities are not handled. It is also not clear whether the manuscript was updated to discuss this.
10. Unclear details and issues with writing / presentation (Wd9S, yC6F)
   - Vagueness in loss function details (Wd9S)
   - Inconsistent terminology (yC6F)
   - Whether the term "coordinate solver" accurately/precisely describes what the 2nd stage does (PrUm)
   - Use of constant 50 for canvas sizing in Algorithm 2 (yC6F)
   - *These were mostly addressed*

Reviewer also had the following concerns which the AC believes to be out-of-scope for this work:
1. No downstream evaluation (yC6F)
- Authors considered use of geometry problem-solving as a downstream application, but note that some problem can be solved symbolically via text only.
2. Limitation to 2D Euclidean Geometry (FpEe, PrUm)

Minor typos / wording:
- L192: "DeepSeeek" => "DeepSeek"
- L964: "In this section, We will supplement necessary experimental analysis (Section 4.2)" => "In this section, we provide experimental analysis for Section 4.2."
- L1023-1025: "The Section 4.2" => "Section 4.2", "figure 14" => "Figure 14", "Section4.2" => "Section 4.2"

**Reviewer Scores:**

Reviewers are mixed on this work with two rejects (Wd9S - reject, FpEe - marginal reject), and two marginal accepts (yC6F, PrUm).

The authors posted a response and revised manuscript (without any markings to indicate what was changed) on Nov 27th, and did not get any follow up from the reviewers.  As the authors did attempt to address some of the reviewer concerns, it is possible that the most negative reviewer may have increased their score to 4 (marginal reject).

---

### Decision · Program_Chairs · 2026-01-26

Reject